# Global diversity and antimicrobial resistance of typhoid fever pathogens: Insights from a meta-analysis of 13,000 *Salmonella* Typhi genomes

Megan E Carey[1,2,3]*, Zoe A Dyson[2,4,5], Danielle J Ingle[6], Afreenish Amir[7], Mabel K Aworh[8,9], Marie Anne Chattaway[10], Ka Lip Chew[11], John A Crump[12], Nicholas A Feasey[13,14], Benjamin P Howden[15,16], Karen H Keddy[17], Mailis Maes[1], Christopher M Parry[13], Sandra Van Puyvelde[1,18], Hattie E Webb[19], Ayorinde Oluwatobiloba Afolayan[20], Anna P Alexander[21], Shalini Anandan[22], Jason R Andrews[23], Philip M Ashton[24,25], Buddha Basnyat[26], Ashish Bavdekar[27], Isaac I Bogoch[28], John D Clemens[29,30,31,32], Kesia Esther da Silva[23], Anuradha De[33], Joep de Ligt[34], Paula Lucia Diaz Guevara[35], Christiane Dolecek[36,37], Shanta Dutta[38], Marthie M Ehlers[39,40], Louise Francois Watkins[19], Denise O Garrett[41], Gauri Godbole[10], Melita A Gordon[42], Andrew R Greenhill[43,44], Chelsey Griffin[19], Madhu Gupta[45], Rene S Hendriksen[46], Robert S Heyderman[47], Yogesh Hooda[48], Juan Carlos Hormazabal[49], Odion O Ikhimiukor[20], Junaid Iqbal[50], Jobin John Jacob[22], Claire Jenkins[10], Dasaratha Ramaiah Jinka[51], Jacob John[52], Gagandeep Kang[52], Abdoulie Kanteh[53], Arti Kapil[54], Abhilasha Karkey[26], Samuel Kariuki[55], Robert A Kingsley[56], Roshine Mary Koshy[57], AC Lauer[19], Myron M Levine[58], Ravikumar Kadahalli Lingegowda[59], Stephen P Luby[23], Grant Austin Mackenzie[53], Tapfumanei Mashe[60,61], Chisomo Msefula[62], Ankur Mutreja[1], Geetha Nagaraj[59], Savitha Nagaraj[63], Satheesh Nair[10], Take K Naseri[64], Susana Nimarota-Brown[64], Elisabeth Njamkepo[65], Iruka N Okeke[20], Sulochana Putli Bai Perumal[66], Andrew J Pollard[67,68], Agila Kumari Pragasam[22], Firdausi Qadri[30], Farah N Qamar[50], Sadia Isfat Ara Rahman[30], Savitra Devi Rambocus[16], David A Rasko[69,70], Pallab Ray[45], Roy Robins-Browne[6,71], Temsunaro Rongsen-Chandola[72], Jean Pierre Rutanga[73], Samir K Saha[48], Senjuti Saha[48], Karnika Saigal[74], Mohammad Saiful Islam Sajib[48,75], Jessica C Seidman[41], Jivan Shakya[76,77], Varun Shamanna[59], Jayanthi Shastri[33,78], Rajeev Shrestha[79], Sonia Sia[80], Michael J Sikorski[58,69,70], Ashita Singh[81], Anthony M Smith[82], Kaitlin A Tagg[19], Dipesh Tamrakar[79], Arif Mohammed Tanmoy[48], Maria Thomas[83], Mathew S Thomas[84], Robert Thomsen[64], Nicholas R Thomson[5], Siaosi Tupua[64], Krista Vaidya[85], Mary Valcanis[16], Balaji Veeraraghavan[22], François-Xavier Weill[65], Jackie Wright[34], Gordon Dougan[1], Silvia Argimón[86], Jacqueline A Keane[1], David M Aanensen[86], Stephen Baker[1,3], Kathryn E Holt[2,4], Global Typhoid Genomics Consortium Group Authorship

*For correspondence:
megan.carey@lshtm.ac.uk

Group author details:
Global Typhoid Genomics
Consortium Group Authorship
See page 32

Competing interest: See page
34

Reviewing Editor: Marc J
Bonten, University Medical
Center Utrecht, Netherlands

[1]Cambridge Institute of Therapeutic Immunology and Infectious Disease (CITIID), University of Cambridge School of Clinical Medicine, Cambridge Biomedical Campus, Cambridge, United Kingdom; [2]Department of Infection Biology, Faculty of Infectious and Tropical Diseases, London School of Hygiene & Tropical Medicine, London, United Kingdom; [3]IAVI, Chelsea & Westminster Hospital, London, United Kingdom; [4]Department of Infectious Diseases, Central Clinical School, Monash University,

Melbourne, Australia; [5]Wellcome Sanger Institute, Wellcome Genome Campus, Hinxton, United Kingdom; [6]Department of Microbiology and Immunology at the Peter Doherty Institute for Infection and Immunity, The University of Melbourne, Melbourne, Australia; [7]National Institute of Health, Islamabad, Pakistan; [8]Nigeria Field Epidemiology and Laboratory Training Programme, Abuja, Nigeria; [9]College of Veterinary Medicine, North Carolina State University, Raleigh, United States; [10]United Kingdom Health Security Agency, London, United Kingdom; [11]National University Hospital, Singapore, Singapore; [12]Centre for International Health, University of Otago, Dunedin, New Zealand; [13]Department of Clinical Sciences, Liverpool School of Tropical Medicine, Liverpool, United Kingdom; [14]Malawi-Liverpool Wellcome Programme, Kamuzu University of Health Sciences, Blantyre, Malawi; [15]Centre for Pathogen Genomics, Department of Microbiology and Immunology, University of Melbourne at Doherty Institute for Infection and Immunity, Melbourne, Australia; [16]Microbiological Diagnostic Unit Public Health Laboratory, The University of Melbourne at the Peter Doherty Institute for Infection and Immunity, Melbourne, Australia; [17]Independent consultant, Johannesburg, South Africa; [18]University of Antwerp, Antwerp, Belgium; [19]Centers for Disease Control and Prevention, Atlanta, United States; [20]Global Health Research Unit (GHRU) for the Genomic Surveillance of Antimicrobial Resistance, Faculty of Pharmacy, University of Ibadan, Ibadan, Nigeria; [21]Lady Willingdon Hospital, Manali, India; [22]Department of Clinical Microbiology, Christian Medical College, Vellore, India; [23]Division of Infectious Diseases and Geographic Medicine, Stanford University, Stanford, United States; [24]Malawi-Liverpool Wellcome Programme, Blantyre, Malawi; [25]Institute of Infection, Veterinary and Ecological Sciences, University of Liverpool, Liverpool, United Kingdom; [26]Oxford University Clinical Research Unit Nepal, Kathmandu, Nepal; [27]KEM Hospital Research Centre, Pune, India; [28]Department of Medicine, Division of Infectious Diseases, University of Toronto, Toronto, Canada; [29]International Vaccine Institute, Seoul, Republic of Korea; [30]International Centre for Diarrhoeal Disease Research, Dhaka, Bangladesh; [31]UCLA Fielding School of Public Health, Los Angeles, United States; [32]Korea University, Seoul, Republic of Korea; [33]Topiwala National Medical College, Mumbai, India; [34]ESR, Institute of Environmental Science and Research Ltd., Porirua, Wellington, New Zealand; [35]Grupo de Microbiologia, Instituto Nacional de Salud, Bogota, Colombia; [36]Centre for Tropical Medicine and Global Health, Nuffield Department of Medicine, University of Oxford, Oxford, United Kingdom; [37]Mahidol Oxford Tropical Medicine Research Unit, Mahidol University, Bangkok, Thailand; [38]ICMR - National Institute of Cholera & Enteric Diseases, Kolkata, India; [39]Department of Medical Microbiology, Faculty of Health Sciences, University of Pretoria, Pretoria, South Africa; [40]Department of Medical Microbiology, Tshwane Academic Division, National Health Laboratory Service, Pretoria, South Africa; [41]Sabin Vaccine Institute, Washington DC, United States; [42]Institute of Infection, Veterinary and Ecological Sciences, University of Liverpool, Liverpool, United Kingdom; [43]Federation University Australia, Churchill, Australia; [44]Papua New Guinea Institute of Medical Research, Goroka, Papua New Guinea; [45]Post Graduate Institute of Medical Education and Research, Chandigarh, India; [46]Technical University of Denmark, Copenhagen, Denmark; [47]Research Department of Infection, Division of Infection and Immunity, University College London, London, United Kingdom; [48]Child Health Research Foundation, Dhaka, Bangladesh; [49]Bacteriologia, Subdepartamento de Enfermedades Infecciosas, Departamento de Laboratorio Biomedico, Instituto de Salud Publica de Chile (ISP), Santiago, Chile; [50]Department of Pediatrics and Child Health, Aga Khan University, Karachi, Pakistan; [51]Rural Development Trust Hospital, Anantapur, India; [52]Department of Community Health, Christian Medical College,

Vellore, India; [53]Medical Research Council Unit The Gambia at London School Hygiene & Tropical Medicine, Fajara, Gambia; [54]All India Institute of Medical Sciences, Delhi, India; [55]Centre for Microbiology Research, Kenya Medical Research Institute, Nairobi, Kenya; [56]Quadram Institute Bioscience, Norwich, United Kingdom; [57]Makunda Christian Hospital, Assam, India; [58]Center for Vaccine Development and Global Health (CVD), University of Maryland School of Medicine, Baltimore, Maryland, USA, Baltimore, United States; [59]Central Research Laboratory, Kempegowda Institute of Medical Sciences, Bengaluru, India; [60]National Microbiology Reference Laboratory, Harare, Zimbabwe; [61]World Health Organization, Harare, Zimbabwe; [62]Kamuzu University of Health Sciences, Blantyre, Malawi; [63]Saint Johns Medical College and Hospital, Bengaluru, India; [64]Ministry of Health, Government of Samoa, Apia, Samoa; [65]Institut Pasteur, Université Paris Cité, Paris, France; [66]Kanchi Kamakoti CHILDS Trust Hospital, Chennai, India; [67]Oxford Vaccine Group, Department of Paediatrics, University of Oxford, Oxford, United Kingdom; [68]The NIHR Oxford Biomedical Research Centre, Oxford, United Kingdom; [69]Department of Microbiology and Immunology, University of Maryland School of Medicine, Baltimore, United States; [70]Institute for Genome Sciences, University of Maryland School of Medicine, Baltimore, United States; [71]Murdoch Children's Research Institute, Royal Children's Hospital, Parkville, Australia; [72]Centre for Health Research and Development, Society for Applied Studies, Delhi, India; [73]University of Rwanda, College of Science and Technology, Kigali, Rwanda; [74]Chacha Nehru Bal Chikitsalaya, Delhi, India; [75]Institute of Biodiversity, Animal Health and Comparative Medicine, University of Glasgow, Glasgow, United Kingdom; [76]Dhulikhel Hospital, Dhulikhel, Nepal; [77]Institute for Research in Science and Technology, Kathmandu, Nepal; [78]Kasturba Hospital for Infectious Diseases, Mumbai, India; [79]Center for Infectious Disease Research & Surveillance, Dhulikhel Hospital, Kathmandu University Hospital, Dhulikhel, Nepal; [80]Research Institute for Tropical Medicine, Department of Health, Muntinlupa City, Philippines; [81]Chinchpada Christian Hospital, Navapur, India; [82]Centre for Enteric Diseases, National Institute for Communicable Diseases, Johannesburg, South Africa; [83]Christian Medical College, Ludhiana, Ludhiana, India; [84]Duncan Hospital, Raxaul, India; [85]University of California Davis, Davis, United States; [86]Centre for Genomic Pathogen Surveillance, Big Data Institute, University of Oxford, Oxford, United Kingdom

## Abstract

**Background:** The Global Typhoid Genomics Consortium was established to bring together the typhoid research community to aggregate and analyse *Salmonella enterica* serovar Typhi (Typhi) genomic data to inform public health action. This analysis, which marks 22 years since the publication of the first Typhi genome, represents the largest Typhi genome sequence collection to date (n=13,000).

**Methods:** This is a meta-analysis of global genotype and antimicrobial resistance (AMR) determinants extracted from previously sequenced genome data and analysed using consistent methods implemented in open analysis platforms GenoTyphi and Pathogenwatch.

**Results:** Compared with previous global snapshots, the data highlight that genotype 4.3.1 (H58) has not spread beyond Asia and Eastern/Southern Africa; in other regions, distinct genotypes dominate and have independently evolved AMR. Data gaps remain in many parts of the world, and we show the potential of travel-associated sequences to provide informal 'sentinel' surveillance for such locations. The data indicate that ciprofloxacin non-susceptibility (>1 resistance determinant) is widespread across geographies and genotypes, with high-level ciprofloxacin resistance (≥3 determinants) reaching 20% prevalence in South Asia. Extensively drug-resistant (XDR) typhoid has become

dominant in Pakistan (70% in 2020) but has not yet become established elsewhere. Ceftriaxone resistance has emerged in eight non-XDR genotypes, including a ciprofloxacin-resistant lineage (4.3.1.2.1) in India. Azithromycin resistance mutations were detected at low prevalence in South Asia, including in two common ciprofloxacin-resistant genotypes.

**Conclusions:** The consortium's aim is to encourage continued data sharing and collaboration to monitor the emergence and global spread of AMR Typhi, and to inform decision-making around the introduction of typhoid conjugate vaccines (TCVs) and other prevention and control strategies.

**Funding:** No specific funding was awarded for this meta-analysis. Coordinators were supported by fellowships from the European Union (ZAD received funding from the European Union's Horizon 2020 research and innovation programme under the Marie Sklodowska-Curie grant agreement No 845681), the Wellcome Trust (SB, Wellcome Trust Senior Fellowship), and the National Health and Medical Research Council (DJI is supported by an NHMRC Investigator Grant [GNT1195210]).

## Editor's evaluation

Although largely descriptive, this meta-analysis of 13,000 published *Salmonella* Typhi (Typhi) genomes is very important to public health. The dataset and presented analysis are convincing, representing the first wholesale analysis of all available Typhi genomes from the last 21 years. The findings are of interest to microbiologists and infectious disease physicians as well as to public health epidemiologists and policy makers, as they are of great significance to tracking the emergence and maintenance of AMR in Typhi and include novel insights into XDR strain emergence in Pakistan, as well as the relationship between MDR maintenance and chromosomal integration.

## Introduction

*Salmonella enterica* serovar Typhi (Typhi) causes typhoid fever, a predominantly acute bloodstream infection associated with fever, headache, malaise, and other constitutional symptoms. If not treated appropriately, typhoid fever can be fatal; mortality ratios are estimated <1% today, but in the pre-antibiotic era ranged from 10% to 20% (*Andrews et al., 2018*; *Stuart and Pullen, 1946*). Historically, the disease was responsible for large-scale epidemics, triggered by the unsanitary conditions created during rapid urbanisation. Typhoid fever has since been largely controlled in many parts of the world due to large-scale improvements in water, sanitation, and hygiene (WASH) (*Cutler and Miller, 2005*), but was still responsible for an estimated 10.9 million illnesses and 116,800 deaths worldwide in 2017, largely in parts of the world where WASH is suboptimal (*GBD 2017 Typhoid and Paratyphoid Collaborators, 2019*). Antimicrobial therapy has been the mainstay of typhoid control, but multidrug resistance (MDR, defined as combined resistance to ampicillin, chloramphenicol, and co-trimoxazole) emerged in the 1970s, and resistance to newer drugs including fluoroquinolones, third-generation cephalosporins, and azithromycin has been accumulating over the last few decades (*Marchello et al., 2020*).

In 2001, the first completed whole genome sequence of Typhi was published (*Parkhill et al., 2001*). The sequenced isolate was CT18, an MDR isolate cultured from a typhoid fever patient in the Mekong Delta region of Vietnam in 1993. The genome was the result of 2 years of work piecing together plasmid-cloned paired-end sequence reads generated by Sanger capillary sequencing. Together with other early bacterial pathogen genomes, including a second Typhi genome (Ty2) published 2 years later in 2003 (*Deng et al., 2003*), the CT18 genome was heralded as a major turning point in the potential for disease control, treatment, and diagnostics, providing new tools for epidemiology, molecular microbiology, and bioinformatics. It formed the basis for new insights into comparative and functional genomics (*Boyd et al., 2003*; *Faucher et al., 2006*), and facilitated early genotyping efforts (*Baker et al., 2008*; *Roumagnac et al., 2006*). When high-throughput sequencing technologies such as 454 and Solexa (subsequently Illumina) emerged, Typhi was an obvious first target for in-depth characterisation of a single pathogen population (*Holt et al., 2008*), and genomics has been increasingly exploited to describe the true population structure and global expansion of this highly clonal pathogen (*Wong et al., 2015*). Now, whole genome sequencing (WGS) is becoming a more routine component of typhoid surveillance. *Salmonella* were among the first pathogens to transition to routine sequencing by public health laboratories in high-income countries (*Chattaway et al., 2019*;

**eLife digest** *Salmonella* Typhi (Typhi) is a type of bacteria that causes typhoid fever. More than 110,000 people die from this disease each year, predominantly in areas of sub-Saharan Africa and South Asia with limited access to safe water and sanitation. Clinicians use antibiotics to treat typhoid fever, but scientists worry that the spread of antimicrobial-resistant Typhi could render the drugs ineffective, leading to increased typhoid fever mortality.

The World Health Organization has prequalified two vaccines that are highly effective in preventing typhoid fever and may also help limit the emergence and spread of resistant Typhi. In low resource settings, public health officials must make difficult trade-off decisions about which new vaccines to introduce into already crowded immunization schedules. Understanding the local burden of antimicrobial-resistant Typhi and how it is spreading could help inform their actions.

The Global Typhoid Genomics Consortium analyzed 13,000 Typhi genomes from 110 countries to provide a global overview of genetic diversity and antimicrobial-resistant patterns. The analysis showed great genetic diversity of the different strains between countries and regions. For example, the H58 Typhi variant, which is often drug-resistant, has spread rapidly through Asia and Eastern and Southern Africa, but is less common in other regions. However, distinct strains of other drug-resistant Typhi have emerged in other parts of the world.

Resistance to the antibiotic ciprofloxacin was widespread and accounted for over 85% of cases in South Africa. Around 70% of Typhi from Pakistan were extensively drug-resistant in 2020, but these hard-to-treat variants have not yet become established elsewhere. Variants that are resistant to both ciprofloxacin and ceftriaxone have been identified, and azithromycin resistance has also appeared in several different variants across South Asia.

The Consortium's analyses provide valuable insights into the global distribution and transmission patterns of drug-resistant Typhi. Limited genetic data were available from several regions, but data from travel-associated cases helped fill some regional gaps. These findings may help serve as a starting point for collective sharing and analyses of genetic data to inform local public health action. Funders need to provide ongoing support to help fill global surveillance data gaps.

*Stevens et al., 2022*), and these systems often capture Typhi isolated from travel-associated typhoid infections, providing an informal mechanism for sentinel genomic surveillance of pathogen populations in typhoid endemic countries (*Ingle et al., 2019*). More recently, WGS has been adopted for typhoid surveillance by national reference laboratories in endemic countries including the Philippines, Nigeria (*Okeke et al., 2022*), and South Africa (*Lagrada et al., 2022*), and PulseNet International is gradually transitioning to WGS (*Davedow et al., 2022*; *Nadon et al., 2017*). Following the first global genomic snapshot study, which included nearly 2000 genomes of Typhi isolated from numerous typhoid prevalence and incidence studies conducted across Asia and Africa (*Wong et al., 2015*), WGS has become the standard tool for characterising clinical isolates. Given the very high concordance between antimicrobial susceptibility to clinically relevant drugs and known genetic determinants of antimicrobial resistance (AMR) in Typhi (*Argimón et al., 2021a*; *Chattaway et al., 2021*; *da Silva et al., 2022*), WGS is also increasingly used to infer resistance patterns.

The adoption of WGS for surveillance relies on the definition of a genetic framework with linked standardised nomenclature, often supplied by multilocus sequence typing (MLST) and core genome multilocus sequence typing (cgMLST) for clonal pathogens. Typhi evolves on the order of 0.5 substitutions per year, much more slowly than host-generalist *Salmonella,* such as *S. enterica* serovars Kentucky and Agona (five substitutions per year) (*Achtman et al., 2021*; *Duchêne et al., 2016*). As a result, the cgMLST approach, which utilises 3002 core genes (*Zhou et al., 2020*) (two-thirds of the genome) and is popular with public health laboratories for analysis of non-typhoidal *S. enterica*, has limited utility for Typhi. Instead, most analyses rely on identifying single nucleotide variants (SNVs) and using these to generate phylogenies. This approach allows for fine-scale analysis of transmission dynamics (although not resolving individual transmission events, due to the slow mutation rate; *Campbell et al., 2018*) and tracking the emergence and dissemination of AMR lineages (*Klemm et al., 2018*; *da Silva et al., 2022*; *Wong et al., 2015*). In the absence of a nomenclature system such as that provided by cgMLST, an alternative strategy was needed for identifying and naming lineages.

To address this challenge, a genotyping framework ('GenoTyphi') was developed that uses marker SNVs to assign Typhi genomes to phylogenetic clades and subclades (*Wong et al., 2016a*), similar to the strategy that has been widely adopted for *Mycobacterium tuberculosis* (*Coll et al., 2014*). The GenoTyphi scheme was initially developed based on an analysis of almost 2000 Typhi isolates from 63 countries (*Wong et al., 2016a*). This dataset was used to define a global population framework based on 68 marker SNVs, which were used to define 4 primary clades, 15 clades, and 49 subclades organised into a pseudo-hierarchical framework. This analysis demonstrated that most of the global Typhi population was highly structured and included many subclades that were geographically restricted, with the exception of Haplotype 58, or H58 (so named by *Roumagnac et al., 2006*, and designated as genotype 4.3.1 under the GenoTyphi scheme). H58 (genotype 4.3.1) was strongly associated with AMR and was found throughout Asia as well as Eastern and Southern Africa (*Wong et al., 2016a*). The GenoTyphi framework has evolved and expanded to reflect changes in global population structure and the emergence of additional AMR-associated lineages (*Dyson and Holt, 2021*), and has been widely adopted by the research and public health communities for the reporting of Typhi WGS data (*Chattaway et al., 2021*; *Ingle et al., 2021*; *da Silva et al., 2022*). The genotyping framework, together with functionality for identifying AMR determinants and plasmid replicons, and generating clustering-based trees, is available within the online genomic epidemiology platform Typhi Pathogenwatch (*Argimón et al., 2021b*). This system is designed to facilitate genomic surveillance and outbreak analysis for Typhi, including contextualisation with global public data, by public health and research laboratories (*Argimón et al., 2021a*; *Ikhimiukor et al., 2022a*; *Lagrada et al., 2022*) without requiring major investment in computational infrastructure or specialist bioinformatics training.

The increasing prevalence of AMR poses a major threat to effective typhoid fever control. The introduction of new antimicrobials to treat typhoid fever has been closely followed by the development of resistance, beginning with widespread chloramphenicol resistance in the early 1970s (*Anderson, 1975*; *Andrews et al., 2018*). By the late 1980s, MDR typhoid had become common. The genetic basis for MDR was a conjugative (i.e. self-transmissible) plasmid of incompatibility type IncHI1 (*Anderson, 1975*), which was first sequenced as part of the Typhi str. CT18 genome in 2001 (*Parkhill et al., 2001*). This plasmid accumulated genes ($bla_{TEM-1}$, *cat*, *dfr,* and *sul*) encoding resistance to all three first-line drugs, mobilised by nested transposons (Tn*6029* in Tn*21*, in Tn*9*) (*Holt et al., 2011b*; *Wong et al., 2015*). The earliest known H58 isolates were MDR, and it has been proposed that selection for MDR drove the emergence and dissemination of H58 (*Holt et al., 2011b*), which is estimated to have originated in South Asia in the mid-1980s (*Carey et al., 2022*; *da Silva et al., 2022*; *Wong et al., 2015*) before spreading throughout South East Asia (*Holt et al., 2011a*; *Pham Thanh et al., 2016b*) and into Eastern and Southern Africa (*Feasey et al., 2015*; *Kariuki et al., 2010*; *Wong et al., 2015*). The MDR transposon has subsequently migrated to the Typhi chromosome on several independent occasions (*Ashton et al., 2015*; *Wong et al., 2015*), allowing for loss of the plasmid and fixation of the MDR phenotype in various lineages. Other MDR plasmids do occur in Typhi but are comparatively rare (*Argimón et al., 2021b*; *Ingle et al., 2019*; *Rahman et al., 2020*; *Tanmoy et al., 2018*; *Wong et al., 2015*).

The emergence of MDR Typhi led to widespread use of fluoroquinolones (mainly ciprofloxacin) as first-line therapy in typhoid fever treatment. Ciprofloxacin non-susceptibility (CipNS, defined by minimum inhibitory concentration [MIC]≥0.06 mg/L) soon emerged and became common, particularly in South and South East Asia (*Chau et al., 2007*; *Dyson et al., 2019*). The genetic basis for this is mainly substitutions in the quinolone resistance determining region (QRDR) of core chromosomal genes *gyrA* and *parC*, which directly impact fluoroquinolone binding. These substitutions have arisen in diverse Typhi strain backgrounds (estimated >80 independent emergences) (*da Silva et al., 2022*) but appear to be particularly common in H58 (4.3.1) subtypes (*Roumagnac et al., 2006*; *da Silva et al., 2022*; *Wong et al., 2015*). The most common genetic pattern is a single QRDR mutation (typically at *gyrA* codon 83 or 87), which results in a moderate increase in ciprofloxacin MIC to 0.06–0.25 mg/L (*Day et al., 2018*) and is associated with prolonged fever clearance times and increased chance of clinical failure when treating with fluoroquinolones (*Pham Thanh et al., 2016a*; *Wain et al., 1997*). An accumulation of three QRDR mutations raises ciprofloxacin MIC to 8–32 mg/L and is associated with higher occurrence of clinical failure (*Pham Thanh et al., 2016a*). Triple mutants appear to be rare, with the exception of a subclade of 4.3.1.2 bearing GyrA-S83F, GyrA-D87N, and ParC-S80I (designated genotype 4.3.1.2.1; *Ingle et al., 2022*), which

emerged in India in the mid-1990s and has since been introduced into Pakistan, Nepal, Bangladesh, and Chile (*Britto et al., 2020*; *Maes et al., 2020*; *da Silva et al., 2022*; *Pham Thanh et al., 2016a*).

The challenge of fluoroquinolone non-susceptible typhoid was met with increased therapeutic use of third-generation cephalosporins (such as ceftriaxone and cefixime) or azithromycin (for non-severe disease) (*Balasegaram et al., 2012*; *Basnyat et al., 2021*; *Rai et al., 2012*). Reports of ceftriaxone treatment failure in late 2016 in Hyderabad, Pakistan, led to the discovery of an extensively drug-resistant (XDR, defined as MDR plus resistance to fluoroquinolones and third-generation cephalosporins) clone of Typhi (genotype 4.3.1.1.P1, a subtype of H58), which subsequently spread throughout Pakistan (*Klemm et al., 2018*; *Rasheed et al., 2020*; *Yousafzai et al., 2019*). This XDR clone harbours a common combination of chromosomal AMR determinants (integrated MDR transposon plus single QRDR mutation, GyrA-83) but has also acquired an IncY-type plasmid carrying resistance genes, including *qnrS* (which, combined with GyrA-83 results in a ciprofloxacin-resistant [CipR] phenotype with MIC >1 mg/L) and the extended spectrum beta-lactamase (ESBL) encoded by *bla*$_{CTX-M-15}$ (*Klemm et al., 2018*). The ESBL gene has subsequently migrated from plasmid to chromosome in some 4.3.1.1.P1 isolates (*Nair et al., 2021*). Other ESBL-producing, ceftriaxone-resistant (CefR) Typhi strains have been identified in India (*Argimón et al., 2021a*; *Jacob et al., 2021*; *Nair et al., 2021*; *Rodrigues et al., 2017*; *Sah et al., 2019*), via both local 'in-country' surveillance and travel-associated infections. The only oral therapy available to treat non-severe XDR Typhi infection is azithromycin (*Levine and Simon, 2018*), which, although effective, shows prolonged bacteremia and fever clearance times in the human challenge model and is not recommended for treatment of complicated typhoid fever (*Jin et al., 2019*). Azithromycin-resistant (AziR) Typhi, which is associated with mutations in the chromosomal gene *acrB*, has now been reported across South Asia (*Carey et al., 2021*; *Duy et al., 2020*; *Iqbal et al., 2020*; *Sajib et al., 2021*) and has been linked to treatment failure in Nepal *Duy et al., 2020*; however, the prevalence so far remains low (*Hooda et al., 2019*; *da Silva et al., 2022*). Imported infections caused by XDR Typhi 4.3.1.1.P1 have been identified in Australia (*Ingle et al., 2021*), Europe (*Herdman et al., 2021*; *Nair et al., 2021*), and North America *Eshaghi et al., 2020*; *François Watkins et al., 2020*; imported AziR Typhi infections are rarer but have been reported in Singapore (*Octavia et al., 2021*).

The accumulation of resistance to almost all therapeutic options means that there is an urgent need to track the emergence and spread of AMR Typhi, both to guide empiric therapy to prevent treatment failure (*Nabarro et al., 2022*), and to direct the deployment of preventative interventions like typhoid conjugate vaccines (TCVs) and WASH infrastructure. Given the wealth of existing and emerging WGS data for Typhi, we aimed to create a system to enhance visibility and accessibility of genomic data to inform current and future disease control strategies, including identifying where empiric therapy may need review, and monitoring the impact of TCVs on AMR and vaccine escape. In forming the Global Typhoid Genomics Consortium (GTGC), we aim to engage with the wider typhoid research community to aggregate Typhi genomic data and standardised metadata to facilitate the extraction of relevant insights to inform public health policy through inclusive, reproducible analysis using freely available and accessible pipelines and intuitive data visualisation. Here, we present a large, geographically representative dataset of 13,000 Typhi genomes, and provide a contemporary snapshot of the global genetic diversity in Typhi and its spectrum of AMR determinants. The establishment of the GTGC, which marked 21 years of typhoid genomics, provides a platform for future typhoid genomics activities, which we hope will inform more sophisticated disease control.

## Methods
### Ethical approvals
Each contributing study or surveillance programme obtained local ethical and governance approvals, as reported in the primary publication for each dataset. For this study, inclusion of data that were not yet in the public domain by August 2021 was approved by the Observational/Interventions Research Ethics Committee of the London School of Hygiene and Tropical Medicine (ref #26408), on the basis of details provided on the local ethical approvals for sample and data collection (*Supplementary file 1*).

## Sequence data aggregation

Attempts were made to include all Typhi sequence data generated in the 20 years since the first genome was sequenced, through August 2021. Genome data and the corresponding data owners were identified from literature searches and sequence database searches (European Nucleotide Archive [ENA]; NCBI Short Read Archive [SRA], and GenBank; Enterobase). Unpublished data, including those from ongoing surveillance studies and routine public health laboratory sequencing, were identified through professional networks, published study protocols (*Carey et al., 2020*), and an open call for participation in the GTGC. All data generators thus identified were invited to join the GTGC and to provide or verify corresponding source information, with year and location isolated being required fields ('metadata', see below). Nearly all those contacted responded and are included as consortium authors on this study. The exceptions, where authors did not respond to email inquiries, were: (i) one genome reported from Malaysia (*Ahmad et al., 2017*) and n=133 draft genomes reported from India (*Katiyar et al., 2020*), which were excluded as sequence reads were not available in NCBI; and (ii) n=39 genomes reported in studies of travel-associated or local outbreaks (*Burnsed et al., 2018*; *Hao et al., 2020*; *Shin et al., 2021*), which were included as raw sequence data and sufficient metadata were publicly available. A further n=850 genomes sequenced by US Centers for Disease Control and Prevention and available in NCBI were excluded from analysis because travel history was unknown and most US cases are travel-associated. *Table 1* summarises all studies and unpublished public health laboratory datasets from which sequence data were sourced.

Whole genome sequence data, in the form of Illumina fastq files, were sourced from the ENA or SRA or were provided directly by the data contributors in the case of data that was unpublished in August 2021. Run, BioSample, and BioProject accessions are provided in *Supplementary file 2*, together with contributed metadata and PubMed or preprint identifiers.

## Sequence analysis

Primary sequence analysis was conducted on the Wellcome Sanger Institute compute cluster. Genotypes, as defined under the GenoTyphi scheme (*Dyson and Holt, 2021*; *Wong et al., 2016a*), were called directly from Illumina reads using Mykrobe v0.12.1 with Typhi typing panel v20221207, and collated using the Python code available at https://github.com/typhoidgenomics/genotyphi (v2.0, doi: 10.5281/zenodo.7430538; *Ingle et al., 2022*).

Illumina reads were assembled using the Centre for Genomic Pathogen Surveillance (CGPS) assembly pipeline v2.1.0 (https://gitlab.com/cgps/ghru/pipelines/dsl2/pipelines/assembly/) (*Underwood, 2020*), which utilises the SPAdes assembler (v3.12.0) (*Bankevich et al., 2012*; *Demin et al., 2020*). One readset failed assembly and was excluded. Assemblies were uploaded to Pathogenwatch to confirm species and serovar, and to identify AMR determinants and plasmid replicons (*Argimón et al., 2021b*). Eight assemblies were excluded as they were identified as non-Typhi: either other serovars of *S. enterica* (2 Paratyphi B, 2 Enteritidis, 1 Montevideo, 1 Newport, 1 Durban) or other species (1 *Klebsiella pneumoniae*). Assemblies >5.5 Mbp or <4.5 Mbp in size were also excluded from further analysis (n=35 excluded, see size distributions in *Figure 1—figure supplement 1*). The resulting 13,000 whole genome assemblies are available in Figshare, doi: 10.26180/21431883 (https://doi.org/10.26180/21431883).

Phylogenetic trees were generated using Pathogenwatch, which estimates pairwise genetic distances between genomes (based on counting SNVs across 3284 core genes) and infers a neighbour-joining tree from the resulting distance matrix (*Argimón et al., 2021b*). The Pathogenwatch collections used to generate the tree files are available at https://bit.ly/Typhi4311P1 (tree showing position of Rwp1-PK1, in context with other genomes from Pakistan) and https://bit.ly/Typhi232 (tree for genotype 2.3.2 genomes).

## Metadata curation and variable definitions

Owners of the contributing studies were asked to provide or update source information relating to their genome data, using a standardised template (http://bit.ly/typhiMeta). Repeat isolates were defined as those that represent the same occurrence of typhoid infection (acute disease or asymptomatic carriage) as one that is already included in the dataset. In such instances, data owners were asked to indicate the 'primary' isolate (either the first, or the best quality, genome for each unique

**Table 1.** Summary of published studies and other data sources.

Details of research studies and public health laboratory data aggregated in this study.

| Published studies PubMed ID or DOI (citation as per reference list) | Total genomes | *Representative cases 2010–2020 | †Travel associated |
|---|---|---|---|
| 11677608 (*Parkhill et al., 2001*) | 1 | 0 | 0 |
| 12644504 (*Deng et al., 2003*) | 1 | 0 | 0 |
| 18660809 (*Holt et al., 2008*) | 4 | 0 | 0 |
| 25392358 (*Hendriksen et al., 2015a*) | 22 | 0 | 0 |
| 25428145 (*Hendriksen et al., 2015b*) | 2 | 0 | 0 |
| 25961941 (*Wong et al., 2015*) | 1736 | 733 | 248 |
| 26411565 (*Baker et al., 2015*) | 30 | 0 | 0 |
| 26974227 (*Pham Thanh et al., 2016a*) | 77 | 77 | 0 |
| 27069781 (*Ashton et al., 2016*) | 489 | 432 | 356 |
| 27331909 (*Pham Thanh et al., 2016b*) | 1 | 1 | 0 |
| 27657909 (*Wong et al., 2016b*) | 128 | 111 | 0 |
| 27703135 (*Wong et al., 2016a*) | 99 | 43 | 43 |
| 28060810 (*Dyson et al., 2017*) | 44 | 0 | 0 |
| 28280021 (*Rodrigues et al., 2017*) | 3 | 0 | 0 |
| 28705963 (*Kong et al., 2017*) | 2 | 0 | 0 |
| 28931025 (*Kuijpers et al., 2017*) | 64 | 59 | 0 |
| 29051234 (*Gul et al., 2017*) | 1 | 0 | 0 |
| 29136410 (*Phoba et al., 2017*) | 1 | 0 | 0 |
| 29216342 (*Day et al., 2018*) | 5 | 4 | 3 |
| 29255729 (*Matono et al., 2017*) | 107 | 0 | 0 |
| 29463654 (*Klemm et al., 2018*) | 100 | 0 | 0 |
| 29616895 (*Djeghout et al., 2018*) | 1 | 0 | 0 |
| 29684021 (*Britto et al., 2018*) | 192 | 169 | 0 |
| 30425150 (*Tanmoy et al., 2018*) | 536 | 0 | 0 |
| 30504848 (*Park et al., 2018*) | 249 | 209 | 0 |
| 30236166 (*Burnsed et al., 2018*) | 30 | 0 | 0 |
| 31225619 (*Oo et al., 2019*) | 39 | 39 | 0 |
| 31513580 (*Ingle et al., 2019*) | 107 | 99 | 91 |
| 31730615 (*Hooda et al., 2019*) | 12 | 0 | 0 |
| 31872221 (*Sah et al., 2019*) | 2 | 0 | 0 |
| 31665304 (*Britto et al., 2020*) | 94 | 94 | 0 |
| 32003431 (*Pragasam et al., 2020*) | 194 | 0 | 0 |
| 32106221 (*Rahman et al., 2020*) | 202 | 147 | 0 |
| 32119918 (*Chirico et al., 2020*) | 1 | 0 | 0 |
| 32217683 (*Tagg et al., 2020*) | 5 | 0 | 0 |
| 32253142 (*Liu et al., 2021*) | 1 | 0 | 0 |
| 32732230 (*Hao et al., 2020*) | 1 | 0 | 0 |

*Table 1 continued on next page*

*Table 1 continued*

| Published studies PubMed ID or DOI (citation as per reference list) | Total genomes | *Representative cases 2010–2020 | †Travel associated |
|---|---|---|---|
| 32883020 (*Rasheed et al., 2020*) | 27 | 27 | 0 |
| 33079054 (*Maes et al., 2020*) | 7 | 7 | 0 |
| 33085725 (*Thanh Duy et al., 2020*) | 116 | 0 | 0 |
| 33347558 (*Mashe et al., 2021*) | 29 | 0 | 0 |
| 34223059 (*Duy et al., 2020*) | 4 | 0 | 0 |
| 33496224 (*Octavia et al., 2021*) | 15 | 15 | 12 |
| 33515460 (*Carey et al., 2021*) | 66 | 66 | 0 |
| 33593966 (*Sajib et al., 2021*) | 80 | 80 | 0 |
| 33651791 (*Shin et al., 2021*) | 8 | 0 | 0 |
| 33704480 (*Nair et al., 2021*) | 58 | 58 | 58 |
| 33965548 (*Jacob et al., 2021*) | 2 | 0 | 0 |
| 34370659 (*Chattaway et al., 2021*) | 631 | 604 | 584 |
| 34463736 (*Gauld et al., 2022*) | 262 | 262 | 0 |
| 34515028 (*Kariuki et al., 2021*) | 136 | 88 | 0 |
| 34529660 (*Guevara et al., 2021*) | 77 | 0 | 0 |
| 34543095 (*Ingle et al., 2021*) | 116 | 116 | 107 |
| 34626469 (*Argimón et al., 2021b*) | 92 | 92 | 0 |
| 34812716 (*Kanteh et al., 2021*) | 16 | 14 | 0 |
| 35344544 (*Dyson et al., 2022*) | 41 | 0 | 0 |
| 35750070 (*da Silva et al., 2022*) | 3402 | 3390 | 0 |
| 35767580 (*Maes et al., 2022*) | 203 | 90 | 0 |
| 35999186 (*Lagrada et al., 2022*) | 190 | 190 | 0 |
| 36026470 (*Ikhimiukor et al., 2022a*) | 22 | 14 | 0 |
| 36094088 (*Sikorski et al., 2022*) | 202 | 174 | 1 |
| 37327220 (*Rutanga et al., 2023*) | 51 | 26 | 0 |
| 37339282 (*Smith et al., 2023*) | 281 | 281 | 13 |
| DOI: 10.1101/2022.09.01.506167 (*Thilliez et al., 2022*) | 57 | 0 | 0 |
| DOI: 10.1101/2022.10.03.510628 (*Carey et al., 2022*) | 463 | 0 | 0 |
| DOI: 10.1101/2023.03.27.23287794 (*Ashton et al., 2023*) | 20 | 20 | 0 |
| DOI: 10.1101/2023.03.11.23286741 (*Dyson et al., 2023*) | 732 | 707 | 0 |
| Previously unpublished public health laboratory data | | | |
| France (Institut Pasteur) | 23 | 23 | 17 |
| New Zealand (ESR) | 99 | 97 | 52 |
| USA (CDC) | 889 | 850 | 712 |
| Total | 13,000 | 9508 | 2297 |

*Table 1 continued on next page*

*Table 1 continued*

| Published studies PubMed ID or DOI (citation as per reference list) | Total genomes | *Representative cases 2010–2020 | †Travel associated |
| --- | --- | --- | --- |

*Genomes associated with assumed acute typhoid cases, isolated from 2010 onwards from non-targeted sampling frames; this is the subset of data used to generate genotype prevalence distributions shown in *Figures 1–3*.

†Genomes recorded as travel-associated and with known travel to a specific country in this region, associated with assumed acute typhoid isolated from 2010 onwards from non-targeted sampling frames.

case) to use in the analysis. Repeat isolates were then excluded from the dataset entirely (excluded from *Supplementary file 2*).

Data provided on the source of isolates (specimen type and patient health status) are shown in *Supplementary file 3*. This information was used to identify isolates that were associated with acute typhoid fever. In total, n=6462 genomes were recorded as isolated from symptomatic individuals. A further n=119 were recorded as isolated from asymptomatic carriers. The remaining genomes had no health status recorded (i.e. symptomatic vs asymptomatic carrier); of these, the majority were isolated from blood (n=3365) or the specimen type was not recorded (n=2522). Since most studies and surveillance programmes are set up to capture acute infections rather than asymptomatic carriers, we defined 'assumed acute illness' genomes as those not recorded explicitly as asymptomatic carriers (n=119) or coming from gallbladder (n=1) or environmental (n=14) samples; this resulted in a total of 12,831 genomes that were assumed to represent acute illness.

We defined 'country of origin' as the country of isolation; or for travel-associated infections, the country recorded as the presumed country of infection based on travel history (*Centers for Disease Control and Prevention, 2011*; *Ingle et al., 2021*; *Ingle et al., 2019*; *Matono et al., 2017*). Countries were assigned to geographical regions using the United Nations Statistics Division standard M49 (see https://unstats.un.org/unsd/methodology/m49/overview/); we used the intermediate region label where assigned, and subregion otherwise. To identify isolate collections that were suitably representative of local pathogen populations, for the purpose of calculating genotype and AMR prevalences for a given setting, data owners were asked to indicate the purpose of sampling for each study or dataset. Options available were either 'Non Targeted' (surveillance study, routine diagnostics, reference lab, other; n=11,086), 'Targeted' (cluster investigation, AMR focused, other; n=1862), or 'Not Provided' (n=17). Only samples from 'Non Targeted' sampling frames with known year of isolation and country of origin were included in national prevalence estimates.

## AMR determinants and definitions

AMR determinants identified in the genome assemblies using Pathogenwatch were used to define AMR genotype as follows. MDR: resistance determinants for chloramphenicol (*catA1* or *cmlA*), ampicillin ($bla_{TEM-1D}$, $bla_{OXA-7}$), and co-trimoxazole (at least one *dfrA* gene and at least one *sul* gene). Ciprofloxacin non-susceptible (CipNS): one or more of the QRDR mutations at GyrA-83, GyrA-87, ParC-80, ParC-84, GyrB-464 or presence of a plasmid-mediated quinolone resistance (PMQR) gene (qnrB, qnrD, qnrS); note, this typically corresponds to MIC ≥0.06 mg/L (*Day et al., 2018*). CipR: QRDR triple mutant (GyrA-83 and GyrA-87, together with either ParC-80 or ParC-84), or PMQR gene together with GyrA-83, GyrA-87, and/or GyrB-464. This typically corresponds to MIC ≥1 mg/L, and CipR is a subset of CipNS. Ceftriaxone resistant (CefR): presence of an ESBL (blaCTX-M-12, blaCTX-M-15, blaCTX-M-23, blaCTX-M-55, blaSHV-12). XDR: MDR plus CipR plus CefR. AziR: mutation at AcrB-717. The above lists all those AMR determinants that were found here in ≥1 genome and used to define AMR profiles and prevalences; additional AMR genes sought by Typhi Pathogenwatch but not detected are listed in Supplementary Table 2 of (*Argimón et al., 2021b*).

## Genotype and AMR prevalence estimates and statistical analysis

All statistical analyses were conducted in R v4.1.2 (*R Development Core Team, 2021*), code is available in R markdown format at https://github.com/typhoidgenomics/TyphoidGenomicsConsortiumWG1 (v1.0, doi:10.5281/zenodo.7487862; *Holt, 2022*). Genotype and AMR frequencies were calculated at the level of country and UN world region (based on 'country of origin') as defined above. Inclusion criteria for these estimates were: known 'country of origin', known year of isolation, non-targeted

sampling, assumed acute illness (see definitions of these variables above). A total of 10,726 genomes met these criteria; the subset of 9478 isolated from 2010 onwards were the focus of the majority of analyses and visualisations, including all prevalence estimates. The prevalence estimates reported in text and figures are simple proportions; 95% confidence intervals (CIs) for proportions are given in text and supplementary tables where relevant. Annual prevalence rates were estimated for countries that had N≥50 representative genomes and ≥3 years with ≥10 representative genomes. Association between MDR prevalence and prevalence of IncHI1 plasmids amongst MDR genomes was assessed for countries with ≥5% MDR prevalence between 2000 and 2020. The significance of increases or decreases in prevalence was assessed using a Chi-squared test for trend in proportions (using the proportion.trend.test function in R). There are no established thresholds for the prevalence of resistance that should trigger changes in empirical therapy recommendations for enteric fever; hence, we defined our own categories of resistance prevalence for visualisation purposes, to reflect escalating levels of concern for empirical antimicrobial use: (i) 0, no resistance detected; (ii) >0 and≤2%, resistance present but rare; (iii) 2–10%, emerging resistance; (iv) 10–50%, resistance common; (v) >50%, established resistance. Robustness of prevalence estimates was assessed informally, by comparing overlap of 95% CIs computed for different laboratories from the same country (for genomes isolated 2010–2020, and laboratories with N≥20 genomes [Southern Asia] or N≥10 [Nigeria] meeting the inclusion criteria during this period).

## Data visualisations

All analyses and plots were generated using R v4.1.2, code is available in R markdown format at https://github.com/typhoidgenomics/TyphoidGenomicsConsortiumWG1 (v1.0, doi:10.5281/zenodo.7487862; *Holt, 2022*). Data processing was done using the R packages tidyverse v1.3.1, dplyr v1.0.7, reshape2 v1.4.4, and janitor v2.1.0; figures were generated using packages ggplot2 v3.3.5, ggExtra v0.9, patchwork v1.1.1, RColorBrewer v1.1-2, and pals v1.7; maps were generated using packages sf v1.0-5, rvest v1.0.2, maps v3.4.0, scatterpie v0.1.7, ggnewscale v0.4.5; trees were plotted using ggtreeio v1.18.1 and ggtree v3.2.1.

# Results

## Overview of available data

A total of 13,000 confirmed Typhi genomes were collated from 65 studies and 5 unpublished public health laboratory datasets (see *Table 1*, *Supplementary file 2*). N=35 genomes had assembly sizes outside of the plausible range (4.5–5.5 Mbp, see *Figure 1—figure supplement 1*), leaving n=12,965 high-quality genomes originating from 110 countries. The distribution of samples by world region (as defined by WHO statistics division M49) is shown in *Table 2*, with country breakdown in *Supplementary file 4*. The majority originated from Southern Asia (n=8231), specifically India (n=2705), Bangladesh (n=2268), Pakistan (n=1810), and Nepal (n=1436). A total of n=1140 originated from South-eastern Asia, with >100 each from Cambodia (n=279), Vietnam (n=224), the Philippines (n=209), Indonesia (n=145), and Laos (n=139). Overall, 1106 genomes originated from Eastern Africa, including >100 each from Malawi (n=569), Kenya (n=254), Zimbabwe (n=110). Other regions of Africa were less well represented, with n=384 from Western Africa, n=317 from Southern Africa, n=59 from Middle Africa (so-named in the M49 region definitions, although more commonly referred to as Central Africa), and n=41 from Northern Africa (see *Table 2* and *Supplementary file 4* for details).

Overall, there were 36 countries with ≥20 genomes (total n=12,409 genomes, 95.7%) and 21 countries with ≥100 genomes (n=11,761 genomes, 90.7%) (see *Supplementary file 4*). Countries with the most genomes available (n≥100 each) were mainly those where local surveillance studies have utilised WGS for isolate characterisation: India (*Britto et al., 2020*; *da Silva et al., 2022*), Bangladesh (*Rahman et al., 2020*; *da Silva et al., 2022*), Nepal (*Britto et al., 2018*; *da Silva et al., 2022*; *Pham Thanh et al., 2016a*), Pakistan (*da Silva et al., 2022*), Cambodia (*Kuijpers et al., 2017*; *Pham Thanh et al., 2016b*), Laos (*Wong et al., 2015*), Vietnam (*Holt et al., 2011a*), Kenya (*Kariuki et al., 2021*; *Kariuki et al., 2010*), Malawi (*Feasey et al., 2015*), Zimbabwe (*Mashe et al., 2021*; *Thilliez et al., 2022*), Ghana (*Park et al., 2018*), Nigeria (*Ikhimiukor et al., 2022a*; *Wong et al., 2016b*), Chile (*Maes et al., 2022*), Samoa *Sikorski et al., 2022*; plus South Africa (*Smith et al., 2023*), the Philippines

**Table 2.** Summary of genomes by region.

| Region | Total genomes | *Representative cases 2010–2020 | †Travel (%) amongst representative cases 2010–2020 |
|---|---|---|---|
| Australia and NZ | 57 | 57 | 0 (0%) |
| Caribbean | 20 | 20 | 20 (100%) |
| Central America | 103 | 100 | 100 (100%) |
| Eastern Africa | 1106 | 830 | 49 (5.9%) |
| Eastern Asia | 12 | 3 | 3 (100%) |
| Eastern Europe | 3 | 1 | 1 (100%) |
| Melanesia | 232 | 37 | 30 (81.1%) |
| Micronesia | 4 | 1 | 1 (100%) |
| Middle Africa | 59 | 21 | 6 (28.6%) |
| Northern Africa | 41 | 6 | 6 (100%) |
| Northern America | 167 | 140 | 2 (1.4%) |
| Northern Europe | 109 | 105 | 0 (0%) |
| Polynesia | 324 | 262 | 45 (17.2%) |
| South America | 367 | 105 | 5 (4.8%) |
| South-eastern Asia | 1140 | 584 | 72 (12.3%) |
| Southern Africa | 317 | 286 | 2 (0.7%) |
| Southern Asia | 8231 | 6623 | 1878 (28.4%) |
| Southern Europe | 10 | 6 | 6 (100%) |
| Western Africa | 384 | 267 | 34 (12.7%) |
| Western Asia | 47 | 21 | 21 (100%) |
| Western Europe | 7 | 3 | 3 (100%) |
| Unknown | 225 | 0 | 0 |
| Total | 12965 | 9478 | 2284 (24.1%) |

*Genomes associated with assumed acute typhoid cases, isolated from 2010 onwards from non-targeted sampling frames; this is the subset of data used to generate genotype prevalence distributions shown in *Figures 1–3*.

†Genomes recorded as travel-associated and with known travel to a specific country in this region, associated with assumed acute typhoid isolated from 2010 onwards from non-targeted sampling frames. Countries were assigned to world regions based on the United Nations (UN) Statistics Division standard M49.

(*Lagrada et al., 2022*), United Kingdom, and United States, where Typhi isolates are sequenced as part of national surveillance programmes.

The genome collection included n=3381 isolates that were recorded as travel-associated (see *Table 2* and *Supplementary file 4*), contributed mainly by public health reference laboratories in England (n=1740), USA (n=749), Australia (n=490), New Zealand (n=144), France (n=116), and Japan (n=104). The most common countries of origin for travel-associated isolates were India (n=1241), Pakistan (n=783), Bangladesh (n=264), Fiji (n=102), Samoa (n=87), Mexico (n=60), Chile (n=49), Papua New Guinea (n=45), Nigeria (n=42), and Nepal (n=39). For some typhoid-endemic countries, the majority of genome data originated from travel-associated infections captured in other countries; those in this category with total n≥10 genomes are Guatemala (n=22/22), El Salvador (n=19/19), Mexico (n=60/61), Peru (n=14/14), Haiti (n=12/12), Morocco (n=12/13), Iraq (n=19/19), Malaysia (n=35/35), Fiji (n=102/144), and Papua New Guinea (n=45/86) (full data in *Supplementary file 4*).

In total, n=10,726 genomes were assumed to represent acute typhoid fever and recorded as derived from 'non-targeted' sampling frames, meaning local population-based surveillance studies or reference laboratory-based national surveillance programmes that could be considered representative

of a given time (year of isolation) and geography (country and region of origin) (see Methods for definitions). The majority of these isolates (n=9478, 88.4%) originate from 2010 onwards; hence, we focus our reporting of genotype and AMR prevalences on this period. Most come from local typhoid surveillance studies (n=5574) or routine diagnostics/reference laboratory referrals capturing locally acquired (n=1543) or travel-associated (n=2284) cases. All prevalence estimates reported in this study derive from this data subset, unless otherwise stated.

## Geographical distribution of genotypes

The breakdown of genotype prevalence by world region, for genomes isolated from non-targeted sampling frames from 2010 onwards, is shown in *Figure 1a* (denominators in *Table 2*, full data in *Supplementary file 5*). Annual breakdown of regional genotype prevalence rates is given in *Figure 1—figure supplement 2* (raw data, proportions, and 95% CIs in *Supplementary file 5*). Notably, while our data confirm that H58 genotypes (4.3.1 and derived) dominate in Asia, Eastern Africa, and Southern Africa, they were virtually absent from other parts of Africa, from South and Central America, as well as from Polynesia and Melanesia (*Figure 1*). Instead, each of these regions was dominated by their own local genotypes. Typhoid fever is no longer endemic in Northern America, Europe, or Australia/New Zealand. The genotype distributions shown for these regions were estimated from Typhi that were isolated locally but not recorded as being travel-associated; nevertheless, these genomes can be assumed to result from limited local transmission of travel-associated infections, and thus to reflect the diversity of travel destinations for individuals living in those regions. Annual national genotype prevalences for well-sampled countries with endemic typhoid are shown in *Figure 1b* (full data in *Supplementary file 6* and *Figure 1—figure supplement 3*). Below, we summarise notable features of the global genotype distribution, by world region (as defined by WHO statistics division, see Methods).

### Southern Asia

Southern Asia was the most represented region, with 6623 genomes suitable for prevalence analysis. The genotype distribution confirms the widely reported finding that the H58 lineage (4.3.1 and derived genotypes) is the dominant form of Typhi in Southern Asia, where it is thought to have originated (*Carey et al., 2022*; *Roumagnac et al., 2006*; *da Silva et al., 2022*; *Wirth, 2015*; *Wong et al., 2015*) (overall prevalence, 70.4% [95% CI, 69.3–71.5%]; n=4662/6623). Notably though, the distribution of H58 genotypes was different between countries in the region (see *Figure 1b*), and in Bangladesh, it was associated with a minority of genomes (42% [n=670/1591], compared with 73% in India [n=1655/2267], 74% in Nepal [n=941/1275], and 94% in Pakistan [n=1390/1484]). India and Nepal were dominated by sublineage 2 (genotype 4.3.1.2 and derived genotypes; 54% [n=1214/2267] and 57% [n=736/1275], respectively), which was rare in Bangladesh (0.6%; n=9/1591) and Pakistan (3.2%; n=47/1484). In India, H58 lineage 1 (4.3.1.1) was also present at appreciable frequency (12%; n=268/2267) as was 4.3.1 (i.e. H58 that does not belong to any of the defined sublineages 4.3.1.1–3; 7.4% [n=168/2267]). In Nepal, 4.3.1 was present at 12% frequency (n=152/1275) and 4.3.1.1 at just 4.9% (n=63/1275).

In Pakistan, lineage 1 (genotype 4.3.1.1 and derived genotypes) was most common (73%; 1089/1484), with the XDR sublineage (genotype 4.3.1.1.P1) appearing in 2016 (*Gul et al., 2017*; *Klemm et al., 2018*; *Rasheed et al., 2020*) and rapidly rising to dominance (87% in 2020 [n=27/31]; see *Figure 1b*). Pakistan also had prevalent 4.3.1 (17%; n=254/1484). H58 lineage 1 (4.3.1.1) was the single most common genotype in Bangladesh, but made up only one-third of the Typhi population (34%; n=546/1591). Bangladesh has its own H58 lineage 3 (4.3.1.3) (*Rahman et al., 2020*; *Tanmoy et al., 2018*), whose prevalence was 7.1% (n=113/1591); only two 4.3.1 isolates and nine 4.3.1.2 isolates were detected. Non-H58 genotypes were also evident in the region, with the greatest diversity in Bangladesh (see *Figure 1b*). Those exceeding 5% in any one country were: 3.3.2 (5.8% in Bangladesh [n=93/1591], 12.9% in Nepal [n=164/1275]), 2.5 in India (8.4%; n=190/2267), 3.3 in India (6.6%; n=150/2267), 2.3.3 in Bangladesh (17.2%; n=274/1591), and 3.2.2 in Bangladesh (6.6%; n=264/1591). Annual prevalence estimates were fairly stable over the past decade, with the exception of the 4.3.1.1.P1 in Pakistan, which emerged in 2016 and became dominant shortly thereafter (see *Figure 1b*).

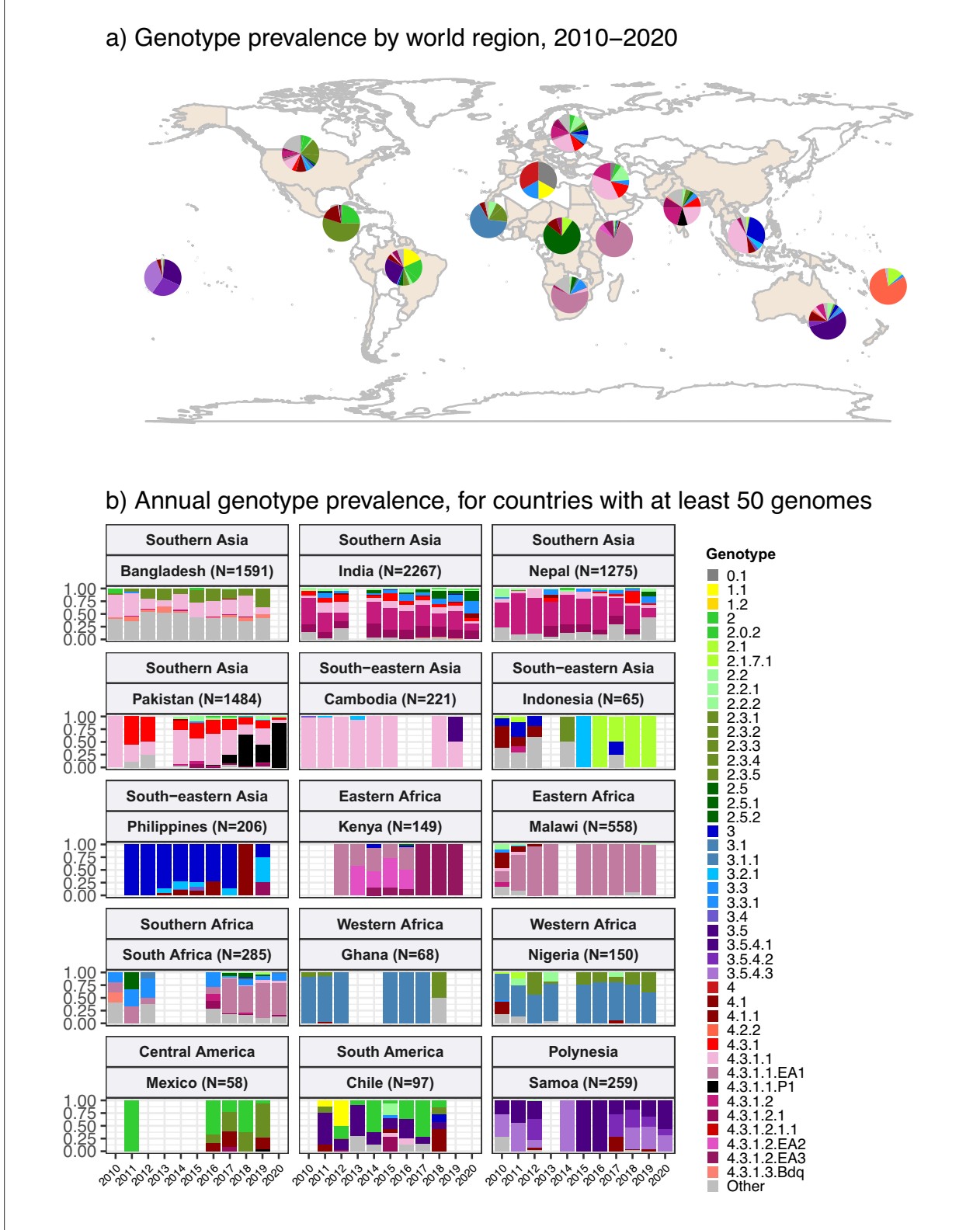

**Figure 1.** Global genotype prevalence estimates. Based on assumed acute cases isolated from untargeted sampling frames from 2010 onwards, with known country of origin (total N=9478 genomes). (**a**) Genotype prevalence by world region, 2010–2020. Countries contributing data are shaded in beige, and are grouped by regions as defined by the UN statistics division. (**b**) Annual genotype prevalence for countries with ≥50 genomes where typhoid is endemic. In both plots, colours indicate prevalence of Typhi genotypes, as per inset legend. Genotypes not exceeding 20% frequency in at least one

*Figure 1 continued on next page*

*Figure 1 continued*

country are aggregated as 'other'. Full data on regional and national genotype prevalences, including raw counts, proportions, and 95% confidence intervals, are given in ***Supplementary files 5 and 6***, respectively.

The online version of this article includes the following figure supplement(s) for figure 1:

**Figure supplement 1.** Genome size pre- and post-filtering, stratified by detection of an IncHI1 plasmid replicon marker.

**Figure supplement 2.** Annual breakdown of genotypes per world region, 2010–2020, for regions with ≥20 representative genomes.

**Figure supplement 3.** Annual breakdown of genotypes per country, for countries with <50 representative genomes between 2010 and 2020.

**Figure supplement 4.** Phylogenetic tree showing relationships amongst genotype 2.3.2 genomes.

## South-eastern and Western Asia

In South-eastern Asia, H58 accounted for 47.3% [95% CI, 43.2–51.3%; 276/584] of isolates in aggregate (mostly 4.3.1.1, 43.0% of total genomes; 251/584). However, the population structures varied between individual countries in the region (see *Figure 1b* and *Figure 1—figure supplement 3*), with H58 accounting for nearly all isolates in Cambodia (98%, n=216/221, all lineage 1), Myanmar (94%, n=46/49, mixed lineages), and Singapore (n=4/4, mixed lineages), but largely absent from Indonesia (3%, n=2/65), Laos (4%, n=1/27), and the Philippines (0.5%, n=1/206). These latter countries showed distinct populations with multiple genotypes exceeding 5% frequency: 4.1 (26%, n=17/65), 3 (18%, n=12/65), 2.1 (15%, n=10/65), and 3.1.2 (12%, n=8/65) in Indonesia; 3.4 (44%, n=12/27), 3.5.2 (15%, n=4/27), 2.3.4 (11%, n=3/27), 3.2.1 (11%, n=3/27), and 4.1 (7%, n=2/27) in Laos; 3 (79%, n=163/206), 3.2.1 (11%, n=23/206), and 4.1 (7%, n=16/206) in the Philippines (*Lagrada et al., 2022*).

Data from Western Asia were limited to a small number of travel-associated infections (total n=21, from Iraq, Lebanon, Qatar, Saudi Arabia, Syria, United Arab Emirates), most of which were H58 (71%; n=15/21); with 38% 4.3.1.1 (n=8/21) and 19% 4.3.1.2 (n=4/21).

## Africa

Only 1410 (15%) of the 9478 genomes from untargeted sampling frames in 2010–2020 were isolated from residents in or travellers to Africa. There is significant underrepresentation from this continent with high endemicity and varying epidemiology across subregions. Our aggregated data confirmed that H58 was the dominant cause of typhoid in Eastern Africa during the study period (93.3% H58 [95% CI, 91.5–95.0%] n=774/830; see *Figure 1a*). It was recently shown that H58 in Kenya was derived from three separate introductions of H58 into the region, which are now assigned their own genotypes (*Kariuki et al., 2021*) (4.3.1.1.EA1, 4.3.1.2.EA2, 4.3.1.2.EA3). Here, we found that at the region level, 4.3.1.1.EA1 dominated (78%, [95% CI 75.1–80.8%] n=647/830; see *Figure 1a*). However, there were country-level differences, with 4.3.1.1.EA1 dominating in Malawi (94%; n=524/558), Tanzania (83%; n=15/18), Zimbabwe (80%; n=20/25), and earlier years in Kenya (59%, n=86/145 in 2012–2016), and 4.3.1.2.EA3 dominating in Rwanda (85%, n=23/27) (*Rutanga et al., 2023*) and Uganda (97%, n=35/36) (*Figure 1b* and *Figure 1—figure supplement 3*). Although the specific periods of sampling differ for these countries, the prevalence of H58 was consistently high across the available time frames for all countries, with no change in dominant genotypes (see *Figure 1b* and *Figure 1—figure supplement 3*; note the apparent shift to 4.3.1.2.EA3 in Kenya is based on n=4 isolates only so requires confirmation).

The majority of Typhi from Southern Africa were isolated in South Africa between 2017 and 2020 (92%; n=262/285), via routine sequencing at the National Institute for Communicable Diseases reference laboratory (*Smith et al., 2023*). H58 prevalence in South Africa was high (69.5%, [95% CI, 63.9–75.1%]; n=182/262) during this time period (mostly 4.3.1.1.EA1, 64%; n=168/262), but was much lower (25% [95% CI, 4–46%]) among the smaller sampling of earlier years (n=4/16 for 2010–2012) (see *Figure 1b*).

In Western Africa, the common genotypes were 3.1.1 (64.4%, [95% CI, 58.7–70.2%]; n=172/266) and 2.3.2 (13.9%, [95% CI, 9.7–18.0%] n=37/266) (*Figure 1a*). Most of these data come from the Typhoid Fever Surveillance in Africa Programme (TSAP) genomics report (*Park et al., 2018*) and a study of typhoid in Abuja and Kano in Nigeria (*Wong et al., 2016b*), which showed that in the period 2010–2013, 3.1.1 dominated in Nigeria and nearby Ghana and Burkina Faso, whereas 2.3.2 dominated in The Gambia and neighbouring Senegal and Guinea Bissau (*Park et al., 2018*). Here, we find that additional data from travel cases and recent Nigerian national surveillance (*Ikhimiukor et al., 2022a*)

suggest that these patterns reflect long-established and persisting populations in the Western African region (see *Figure 1b* and *Figure 1—figure supplement 3*): 3.1.1 was detected from Benin (2002–2009; n=4/4), Burkina Faso (2006–2013; n=11/17), Cote d'Ivoire (2006–2008; n=4/4), The Gambia (2015; n=2/28), Ghana (2007–2017; n=93/109), Guinea (2009; n=1/2), Mali (2008; n=1/5), Mauritania (2009; n=1/2), Nigeria (2008–2019; n=122/192), Sierra Leone (2015–2017; n=2/2), and Togo (2004–2006; n=2/3); and 2.3.2 from Burkina Faso (2012–2013; n=2/17), The Gambia (2008–2014; n=25/28), Ghana (2010–2018; n=9/109), Guinea Bissau (2012–2013; n=2/3), Mali (1999–2018; n=3/5), Niger (1990–1999; n=2/4), Nigeria (1984–2002; n=4/192), Senegal (2012; n=6/10), and Togo (2001; n=1/3).

Very limited genome data were available from the Middle Africa region (n=19; *Table 2*). Genomes from Democratic Republic of the Congo (DRC) comprised 16 genotype 2.5.1 isolates (15 isolated locally, plus one from USA CDC) and a single 4.3.1.2.EA3 isolate (from the UK reference lab). Two genomes each were available from Angola (both 4.1.1, via UK) and Chad (both 2.1, via France). Northern Africa was similarly poorly represented, with one isolate from Egypt (0.1, via UK), two from Morocco (0.1, via UK and 1.1, via USA), two from Sudan (genotype 4, via UK), and one from Tunisia (3.3, from UK).

## The Americas

Strikingly, Central American isolates were dominated by 2.3.2 (55%, [95% CI, 45.2–64.8%] n=55/100), which was also common in Western Africa (13.9%, [95% CI, 9.7–18.0%]; n=37/266) (*Figure 1a*). Little has been reported about Typhi populations from this region previously, and the genomes collated here were almost exclusively novel ones contributed via the US CDC and isolated between 2016 and 2019. The available genomes for the period 2010–2020 mainly originated from El Salvador (n=19, 2012–2019, 89% 2.3.2), Guatemala (n=22, 2016–2019, 41% 2.3.2), and Mexico (n=58, 2011–2019, 50% 2.3.2). Prior to 2010, genotype 2.3.2 was also identified in isolates from Mexico referred to the French reference lab in 1972 (representing a large national outbreak; *Baine et al., 1977*) and 1998. The distance-based phylogeny for 2.3.2 included several discrete clades from different geographical regions in West Africa and the Americas (see *Figure 1—figure supplement 4*), consistent with occasional continental transfers between these regions followed by local clonal expansions. Three clades were dominated by West African isolates (one with isolates from West Coast countries, and two smaller clades from Nigeria and neighbouring countries); two clades of South American isolates (from Chile, Argentina, and Peru); one small clade of Caribbean (mainly Haiti) and USA isolates; and one large clade of Central American isolates (from Mexico, Guatemala, and El Salvador) (see *Figure 1—figure supplement 4*). Other common genotypes identified in Central America were 2.0.2 (overall prevalence 24% [95% CI, 16–32%, n=24/100]; 32% in Guatemala [n=7/22], 26% in Mexico [n=15/58], 11% in El Salvador [n=2/19]) and 4.1 (17%, [95% CI, 9.6–24% n=17/100]; 23% in Guatemala [n=5/22], 21% in Mexico [n=12/58], not detected from El Salvador).

There were 105 genomes available from South America, of which 92% (n=97) were from a recent national surveillance study in Chile (*Maes et al., 2022*). South American Typhi were genetically diverse, with no dominant genotype accounting for the majority of cases in the 2010–2020 period (*Figure 1a*). Genotypes with ≥5% prevalence in the region were 3.5 (27%; n=28/105), 1.1 (18%; n=19/105), 2 (18%; n=19/105), 1.2.1 (5.7%; n=6/105), and 2.0.2 (5.7%; n=6/105). WGS data recently reported by Colombia's Instituto Nacional de Salud (*Guevara et al., 2021*) were not included in the regional prevalence estimates as they covered only a subset (5%) of surveillance isolates that were selected to maximise diversity, rather than to be representative. However, only four genotypes were detected in the Colombia study (1.1, 2, 2.5, 3.5), and two-thirds of isolates sequenced were genotype 2.5 (67%; n=51/77); 3.5 was also common, at 25% (n=20/77) (*Guevara et al., 2021*). Similarly, all five isolates from French Guiana (sequenced via the French reference laboratory) were genotype 2.5, consistent with limited diversity and a preponderance of genotype 2.5 organisms in the north of the continent.

## Pacific Islands

In Melanesia and Polynesia, each island has their own dominant genotype (*Figure 1a*): 2.1.7 and its derivatives in Papua New Guinea (n=5/5 in post-2010 genomes, consistent with the longer-term trend) (*Dyson et al., 2022*), 3.5.3 and 3.5.4 in Samoa (96%; n=249/259, consistent with a recent report) (*Sikorski et al., 2022*), and 4.2 and its derivatives in Fiji (97%; n=31/32, consistent with recent data that was not yet available at the time of this analysis) (*Davies et al., 2022*).

## Global distribution of AMR

We estimated the regional and national prevalence of clinically relevant AMR profiles in Typhi for the period 2010–2020, inferred from WGS data from non-targeted sampling frames for which country of origin could be determined (as per genotype prevalences, see Methods). In order to understand the potential implications of these AMR prevalences for local empirical therapy, we categorised them according to a traffic light-style system (see Methods), whereby amber colours signal emerging resistance of potential concern (<10%), and red colours signal prevalence rates of AMR that may warrant reconsideration of empirical antimicrobial use (>10%; see *Figure 2* and *Figure 2—figure supplement 1*). The regional view (*Figure 2—figure supplement 1*, *Supplementary file 7*) highlights that CipNS is widespread, whereas CipR, AziR, and XDR have been mostly restricted to Southern Asia. MDR was most prevalent in African regions, and to a lesser degree in Asia. Full country-level data is mapped in *Figure 2—figure supplement 2* and detailed in *Supplementary file 8*. National estimates for countries with sufficient data where typhoid is endemic (≥50 representative genomes available for the period 2010–2020, see *Figure 2*) indicate that MDR remains common across all well-sampled African countries (39% in Nigeria, 61% in South Africa, 66% in Ghana, 78% in Kenya, 93% in Malawi), but is much more variable in Asia (3% in India [n=67/2267] and Nepal [n=36/1275], 25% in Bangladesh [n=393/1591], 68% in Pakistan [n=1004/1484], 76% in Cambodia [n=167/221]) and essentially absent from Indonesia (n=0), the Philippines (n=0), Samoa (n=0), Mexico (n=1, 1.7%), and Chile (n=0). The underlying genotypes are shown in *Figure 2—figure supplement 3*, and highlight that MDR in Asia, Eastern Africa, and Southern Africa has been mostly associated with H58 (i.e. 4.3.1 and derived genotypes) but in Western Africa is associated with the dominant genotype in that region, 3.1.1. In contrast, CipNS was associated with more diverse Typhi genotypes in each country, including essentially all common genotypes in Southern Asian countries (*Figure 2—figure supplement 3*). National annual prevalence data suggest that AMR profiles were mostly quite stable over the last decade (with the notable exception of the emergence and rapid spread of XDR Typhi in Pakistan) but reveal some interesting differences between settings in terms of AMR trends and the underlying genotypes (see *Figure 3*, *Figure 2—figure supplement 3*, *Figure 2—figure supplement 4*, *Figure 3—figure supplement 1*).

### Ciprofloxacin non-susceptible

CipNS was near-ubiquitous (exceeding 95% prevalence) in India and Bangladesh throughout the period 2010–2020 (*Figure 3*, *Figure 3—figure supplement 1*). This was associated mainly with GyrA-S83F (79% prevalence in Bangladesh, 70% in India) and GyrA-S83Y mutations (9.2% prevalence in Bangladesh, 26% in India), which were detected across diverse genotype backgrounds (*Figure 2—figure supplement 3*, *Figure 2—figure supplement 4*); in total, CipNS variants were present in 30 genotype backgrounds in India (out of n=34 genotypes, 88%) and 17 in Bangladesh (out of n=21 genotypes, 81%). In neighbouring Nepal, CipNS prevalence has stabilised in the 85–95% range since 2011 (70% GyrA-S83F, 12% GyrA-S83Y; CipNS in 12 genotype backgrounds) (see *Figure 3* and *Figure 2—figure supplement 3*). The persistence of ciprofloxacin-susceptible Typhi in Nepal was largely associated with genotype 3.3.2, which maintained annual prevalence of 3–10% (mean 5.8%) throughout 2010–2018, rising to 39% in 2019. In Pakistan, CipNS has exceeded 95% since 2012 (*Figure 3*), across n=14/17 genotypes (*Figure 2—figure supplement 3*). Sustained high prevalence of CipNS was also evident in Cambodia (4.3.1.1 with GyrA-S83F). In contrast, CipNS has been relatively rare in African countries, but has been increasing in recent years, especially in Kenya (from 20% in 2012 to 65% in 2016, $p=3 \times 10^{-9}$ using proportion trend test) and Nigeria (from 8% in 2013 to 80% in 2019, $p=7 \times 10^{-6}$; see *Figure 3*). CipNS in these settings was associated with QRDR mutations in the locally dominant genotypes, specifically GyrA-S83F (15% of 4.3.1.1.EA1), GyrA-S83Y (100% of 4.3.1.2.EA3) and GyrA-S464F in Kenya (100% of 4.3.1.2.EA2), and GyrA-S83Y (27% of 3.1.1) in Nigeria (see *Figure 2—figure supplement 3* and *Figure 2—figure supplement 4*).

### Ciprofloxacin resistant

CipR emerges in a stepwise manner in Typhi, through acquisition of additional QRDR mutations and/or PMQR genes in strains already carrying a QRDR mutation. CipR genomes were common (≥10%) in Pakistan, India, and Nepal, and emerging (3–6%) in Bangladesh, South Africa, Chile, and Mexico (*Figure 2*). A total of 26 distinct CipR genotypes (comprising unique combinations of Typhi genotype,

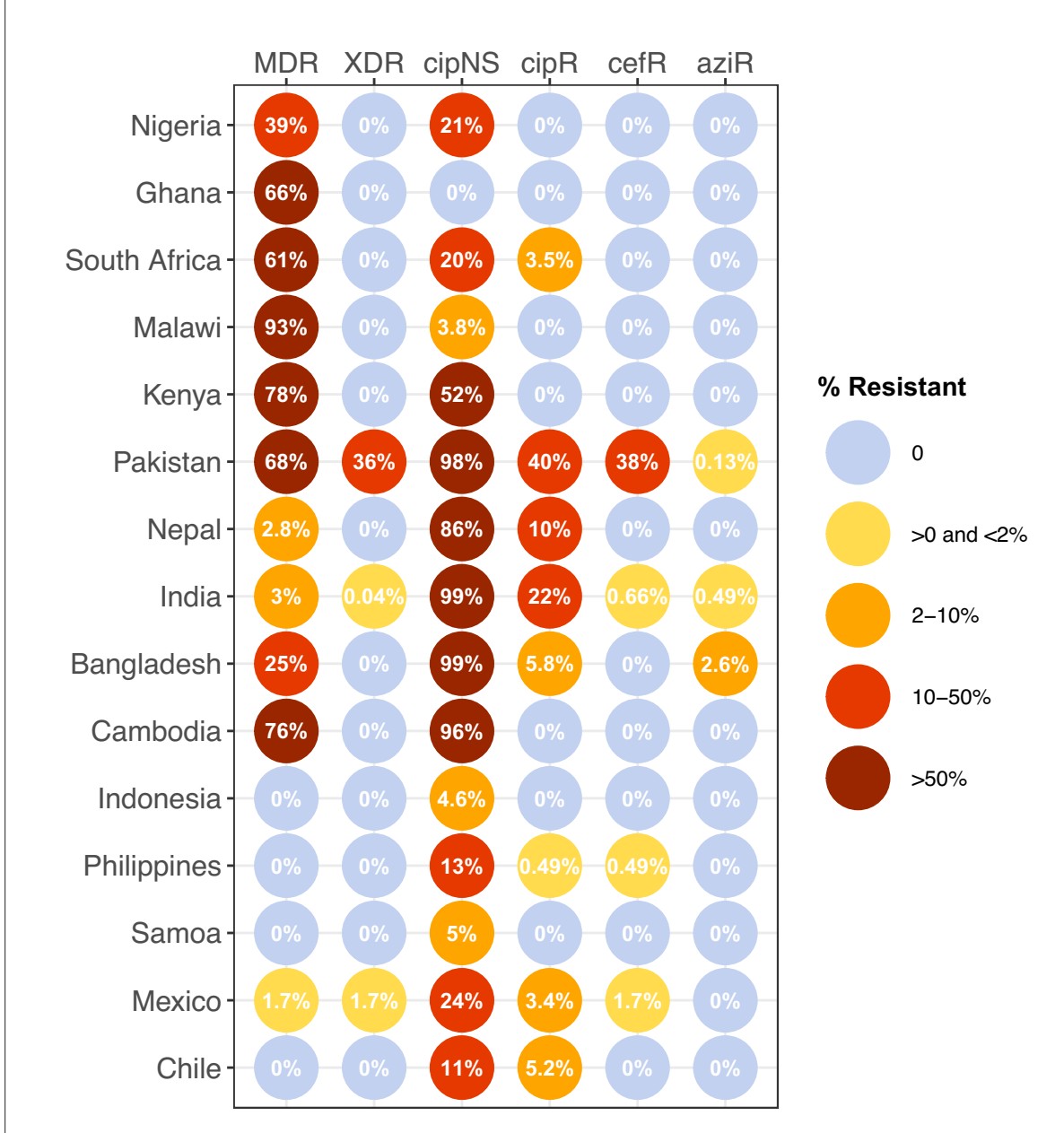

**Figure 2.** Prevalence of key antimicrobial resistance (AMR) genotype profiles by country. For all countries with ≥50 representative genomes (untargeted, assumed acute cases) from 2010 to 2020, where typhoid is endemic. Percentage resistance values are printed for each country/drug combination, and are coloured by categorical ranges to reflect escalating levels of concern for empirical antimicrobial use: (i) 0: no resistance detected; (ii) >0 and ≤2%: resistance present but rare; (iii) 2–10%: emerging resistance; (iv) 10–50%: resistance common; (v) >50%: established resistance. Annual rates underlying these summary rates are shown in **Figure 3** and **Supplementary file 8**. Full data including counts and confidence intervals are included in **Supplementary file 8**. MDR, multidrug resistant; XDR, extensively drug resistant; CipNS, ciprofloxacin non-susceptible; CipR, ciprofloxacin resistant; CefR, ceftriaxone resistant; AziR, azithromycin resistant. Countries are grouped by geographical region.

The online version of this article includes the following figure supplement(s) for figure 2:

**Figure supplement 1.** Prevalence of key antimicrobial resistance (AMR) genotype profiles by world region, for non-targeted samples, 2010–2020.

**Figure supplement 2.** Antimicrobial resistance (AMR) prevalence for non-targeted samples, 2010–2020.

**Figure supplement 3.** Annual genotype prevalence amongst multidrug-resistant (MDR) and ciprofloxacin non-susceptible (CipNS) genomes.

**Figure supplement 4.** Distribution of fluoroquinolone resistance determinants by genotype.

**Figure supplement 5.** Ciprofloxacin-resistant genotypes identified.

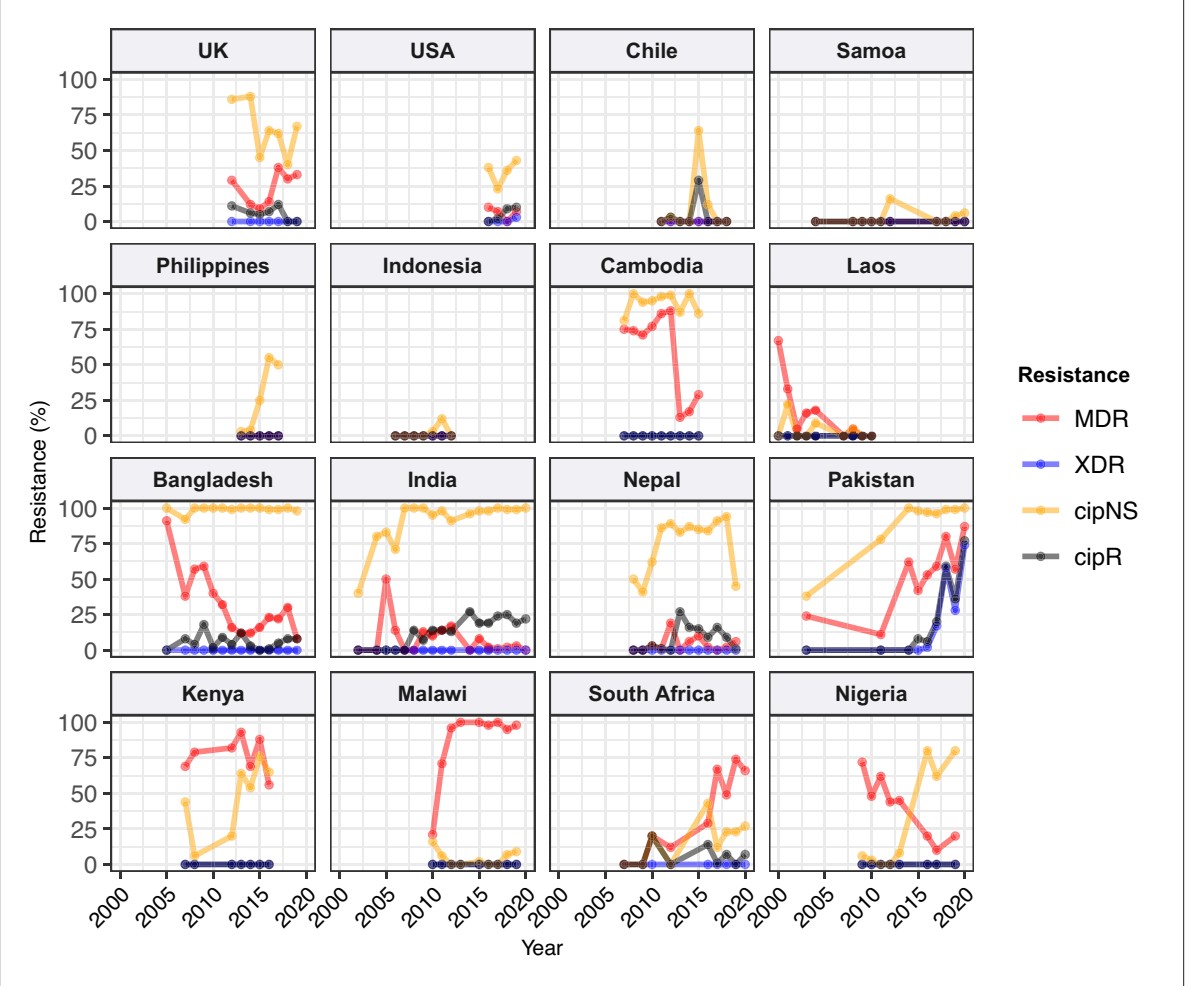

**Figure 3.** Annual prevalence of key antimicrobial resistance (AMR) profiles. For countries with ≥3 years with ≥10 representative genomes (untargeted, assumed acute cases) from 2000 to 2020. Data are shown only for country/year combinations with N≥5 isolates. MDR, multidrug resistant; XDR, extensively drug resistant; CipNS, ciprofloxacin non-susceptible; CipR, ciprofloxacin resistant.

The online version of this article includes the following figure supplement(s) for figure 3:

**Figure supplement 1.** Annual prevalence of key antimicrobial resistance (AMR) profiles.

**Figure supplement 2.** Trends in annual frequency of multidrug-resistant (MDR) genomes and proportion of MDR explained by IncHI1 plasmids.

QRDR mutations, and/or PMQR genes) were identified, of which five were found in appreciable numbers (>5 genomes each, see *Figure 2—figure supplement 5*). The XDR strain 4.3.1.1.P1 (carrying GyrA-S83*F*+*qnrS*) was first identified in Pakistan in 2016 (*Klemm et al., 2018*; *Rasheed et al., 2020*), and here accounted for 75% of Typhi genomes from Pakistan in 2020 and a dramatic rise in CipR prevalence (*Figure 3*). This genotype was only detected three times without a known origin in Pakistan (one isolate each in India, Mexico, and USA, see *Figure 2—figure supplement 5*). The CipR strain 4.3.1.3.Bdq (carrying GyrA-S83F and *qnrS*) emerged in Bangladesh in ~1989 (*da Silva et al., 2022*) and here accounted for 95% of CipR genomes in this country. 4.3.1.3.Bdq genomes were also detected in India (n=4), Singapore (n=1), and South Africa (n=1). The other major CipR genotypes were the QRDR triple-mutant 4.3.1.2.1, its derivative 4.3.1.2.1.1 (which also carries plasmid-borne *qnrB*), and a QRDR triple-mutant sublineage of 3.3. These three CipR variants were most common in India, where we estimated consistently high CipR prevalence (19–27% per year) from 2014 onwards (*Figure 3*), associated with 15 unique CipR genotypes (*Figure 2—figure supplement 5*). Most Indian CipR genomes belong to 4.3.1.2.1 (92.3%). CipR 4.3.1.2.1 was also found in 12 other countries, most notably Nepal (accounting for 95% of CipR genomes), where it has been shown to have been introduced from India and result in treatment failure (*Pham Thanh et al., 2016a*); Pakistan (accounting for

6.6% of CipR genomes); Myanmar (accounting for n=17/17 CipR genomes); and Chile (accounting for n=5/5 CipR genomes) (see *Figure 2—figure supplement 5*). The 3.3 QRDR triple-mutant accounted for 3.8% of CipR genomes in India, and was also found in neighbouring Nepal (n=4, 3% of CipR). CipR genomes were identified from Zimbabwe (4.3.1.1.EA1 with gyrA S83F+*qnrS*, associated with recent CipR outbreaks; *Thilliez et al., 2022*) and South Africa (five different genotypes, totalling 3.5%; see *Figure 2—figure supplement 5*; *Smith et al., 2023*), but were otherwise absent from African Typhi genomes.

## Multidrug resistant

Prevalence of MDR (co-resistance to ampicillin, chloramphenicol, and co-trimoxazole) has declined in India (p=2 × 10⁻⁹ using proportion trend test) to 2% (0–3% per year, 2016–2020), and is similarly rare in Nepal (mean 5% in 2011–2019) (see *Figure 3*). MDR prevalence has also declined in Bangladesh (p=2 × 10⁻⁴ using proportion trend test) but remains high enough to discourage deployment of older first-line drugs, with prevalence exceeding 20% in most years (see *Figure 3*). In Pakistan, the emergence of the XDR strain 4.3.1.1.P1 has driven up MDR prevalence dramatically (p=4 × 10⁻¹¹ using proportion trend test), to 87% in 2020 (see *Figure 3* and *Figure 2—figure supplement 3b*). MDR prevalence has remained high in Kenya and Malawi since the first arrival of MDR H58 strains (estimated early 1990s in Kenya [*Kariuki et al., 2021*]; 2009 in Malawi [*Feasey et al., 2015*]), but has declined steadily in Nigeria, from 72% in 2009 to 10% in 2017 (p=3 × 10⁻⁴ using proportion trend test; see *Figure 3*). All MDR isolates in Nigeria were genotype 3.1.1 and carried large IncHI1 MDR plasmids, which are associated with a fitness cost (*Doyle et al., 2007*). Chromosomal integration of the MDR transposon, which accounted for 100% of MDR in Malawi and 19% in Kenya (all in H58 genotype backgrounds), is associated with comparably lower fitness cost; and this difference in fitness cost may explain why MDR has remained at high prevalence in some settings (where resistance is chromosomally integrated) while declining in other settings (where resistance is plasmid-borne).

*Figure 3—figure supplement 2* shows prevalence of MDR overlaid with prevalence of IncHI1 plasmid carriage amongst MDR strains. Two countries showed a significant rise in MDR prevalence (Pakistan, p=4 × 10⁻¹¹; South Africa, p=9 × 10⁻⁸); in both countries, this rise coincided with loss of IncHI1 plasmids (see *Figure 3—figure supplement 2*) and assumed migration of MDR to the chromosome (as has been clearly shown in XDR 4.3.1.1.P1 strains in Pakistan) (*Klemm et al., 2018*). A decline in the prevalence of MDR over time was observed in Cambodia as in Nigeria, whereby all MDR strains belonged to the same genotype (4.3.1.1 in Cambodia, 3.1.1 in Nigeria) and carried the IncHI1 plasmid (see *Figure 3—figure supplement 2*). As noted above, MDR was maintained at high prevalence rates in Kenya and Malawi, where the IncHI1 plasmid frequency was either in decline (Kenya) or entirely absent (Malawi; see *Figure 3—figure supplement 2*). Notably, a significant decline in total MDR prevalence was observed in Bangladesh (p=2 × 10⁻⁴), and in MDR prevalence within the dominant genotype 4.3.1.1 (p=0.049), despite the majority of MDR (and all MDR within 4.3.1.1) being chromosomal rather than plasmid-associated (*Rahman et al., 2020*; *da Silva et al., 2022*). However, as noted above, MDR did persist in Bangladesh (exceeding 20% prevalence in most years). This is consistent with the hypothesis that the MDR plasmid is associated with a fitness cost that is removed when the MDR transposon becomes chromosomally integrated.

## Extensively drug resistant

The XDR 4.3.1.1.P1 sublineage (i.e. MDR with additional resistance to fluoroquinolones and third-generation cephalosporins including ceftriaxone) was recognised as emerging in late 2016 in Sindh Province, where it caused an outbreak of XDR typhoid that has since spread throughout Pakistan (*Klemm et al., 2018*; *Nair et al., 2021*; *Rasheed et al., 2020*). Here, we identified the genome of strain Rwp1-PK1 (assembly accession NIFP01000000), isolated from Rawalpindi in July 2015, as genotype 4.3.1.1.P1. Rwp1-PK1 was isolated from a 17-year-old male with symptomatic typhoid whose infection did not resolve following ceftriaxone treatment and was found to be phenotypically XDR (resistant to ampicillin, co-trimoxazole, chloramphenicol, ciprofloxacin, ceftriaxone) (*Munir et al., 2016*). The isolate was later sequenced and reported as carrying *bla*CTX-M-15, *bla*TEM-1, *qnrS1*, and GyrA-S83F (*Gul et al., 2017*), but was not genotyped nor included in comparative genomics analyses investigating the emergence of XDR in Pakistan, so has not previously been recognised as belonging to the 4.3.1.1.P1 XDR sublineage. We found that the Rwp1-PK1 genome carries the 4.3.1.1.P1 marker

SNV, clusters with the 4.3.1.1.P1 sublineage in a core-genome tree (*Figure 4*), and shares the full set of AMR determinants typical of 4.3.1.1.P1, indicating that this XDR strain was present in northern Pakistan for at least a full year before it was reported as causing outbreaks in the southern province of Sindh.

## Ceftriaxone resistant

There was no evidence for establishment of 4.3.1.1.P1 nor other XDR lineages outside Pakistan. However, ESBL genes were identified in n=32 non-4.3.1.1.P1 genomes, belonging to eight other genotypes (*Table 3*). Several carried a $bla_{CTX-M-15}$; these include instances with no other acquired AMR genes (genotype 3 in the Philippines [*Hendriksen et al., 2015b*; *Lagrada et al., 2022*]; genotype 4.3.1.2 in Iraq [*Nair et al., 2021*]); one instance with chromosomally integrated AMR genes plus IncY plasmid-borne $bla_{CTX-M-15}$ (genotype 2.5.1 in DRC; *Phoba et al., 2017*); and instances with a 4.3.1.1.P1-like profile carrying *qnrS* in the IncY plasmid and the MDR locus in the chromosome (n=4 4.3.1, India and Pakistan; n=1 4.3.1.1, Pakistan; see *Table 3*). However, overall, $bla_{CTX-M-15}$ IncY plasmids were rare (n=1–4 genomes) in all genotype backgrounds except 4.3.1.1.P1 (total n=655), suggesting that the IncY $bla_{CTX-M-15}$ plasmid has not been stably maintained in other Typhi lineages (see *Table 3*). IncY plasmids were also identified in a single genotype 2.3.3 organism isolated in the UK in 1989 associated with travel to Pakistan (carrying *catA1, tetA(B)*); and in a sublineage of IncHI1-negative 3.1.1 genomes from Nigeria carrying $bla_{TEM-1D}$, *dfrA14, sul2, tetA(A)*, as has been recently reported (*Ikhimiukor et al., 2022a*; *Wong et al., 2016b*). Other examples of ESBL carriage in Typhi genomes appear to represent isolated events (1 or 2 genomes per ESBL/plasmid or ESBL/genotype combination, see *Table 3*), except for a sublineage of 4.3.1.2.1 from India carrying $bla_{SHV-12}$ in a IncX3 plasmid backbone. Concerningly, the plasmid also carries *qnrB* and is present in the well-established 4.3.1.2.1 QRDR triple-mutant strain background, resulting in a combination of resistance to ciprofloxacin, third-generation cephalosporins and ampicillin (*Argimón et al., 2021b*; *Chattaway et al., 2021*; *Ingle et al., 2021*; *Jacob et al., 2021*) (although lacking resistance determinants for chloramphenicol, co-trimoxazole, and azithromycin). This group comprised 15 isolates from Mumbai (*Argimón et al., 2021b*; *Jacob et al., 2021*) (across two studies, 2015–2018), plus three additional isolates from travellers returning to England, Australia, and the USA from India (*Chattaway et al., 2021*; *Ingle et al., 2021*) (2018–2020). This strain therefore appears to have originated in Mumbai and persisted there since at least 2015 for at least 6 years, but our data do not indicate onward spread out of Maharashtra or India.

## Azithromycin resistant

AziR-associated mutations in *acrB* were identified in 74 genomes. The majority of *acrB* mutants were from Bangladesh (n=55, 73%), followed by India (n=11, 15%) (see *Figure 5a*), although the overall prevalence of resistance was very low even in these locations (2.6% in Bangladesh, 0.5% in India). Thirteen distinct combinations of genotype and *acrB* mutation were identified, implying at least thirteen independent events of AziR emergence; six were singleton isolates, and four were represented by two to three isolates each (*Figure 5b*). The three more common AziR variants all carried R717Q, in 4.3.1.1 (n=38, mainly from Bangladesh), 3.2.2 (n=12, from Bangladesh), or 4.3.1.2 (n=7, from India). Notably, half (n=7/13) of all *acrB*/genotype combinations were identified in Bangladesh (see *Figure 4b*). All *acrB* mutants also carried QRDR mutations, and eight were cipR: n=6 belong to the CipR 4.3.1.2.1 lineage in India (all carried R717Q and were isolated in 2017 in Chandigarh) and n=2 belong to the CipR 4.3.1.3.Bdq lineage (both carried R717L and were isolated in 2019, one in Singapore and one in Bangladesh).

## Robustness of national estimates across studies

The estimates of genotype and AMR prevalence represented here reflect post hoc analyses of data that were generated for a variety of different primary purposes in different settings, by different groups using varied criteria for sample collection, including in-country surveillance and travel-associated cases recorded in other countries. Whilst datasets known to be biased towards sequencing of AMR strains including outbreak investigations were excluded from prevalence estimates, there is still substantial heterogeneity across data sources. To explore the robustness of these national-level estimates, we compared prevalence estimates for the same country from different studies/sources, where sufficient data existed to do so.

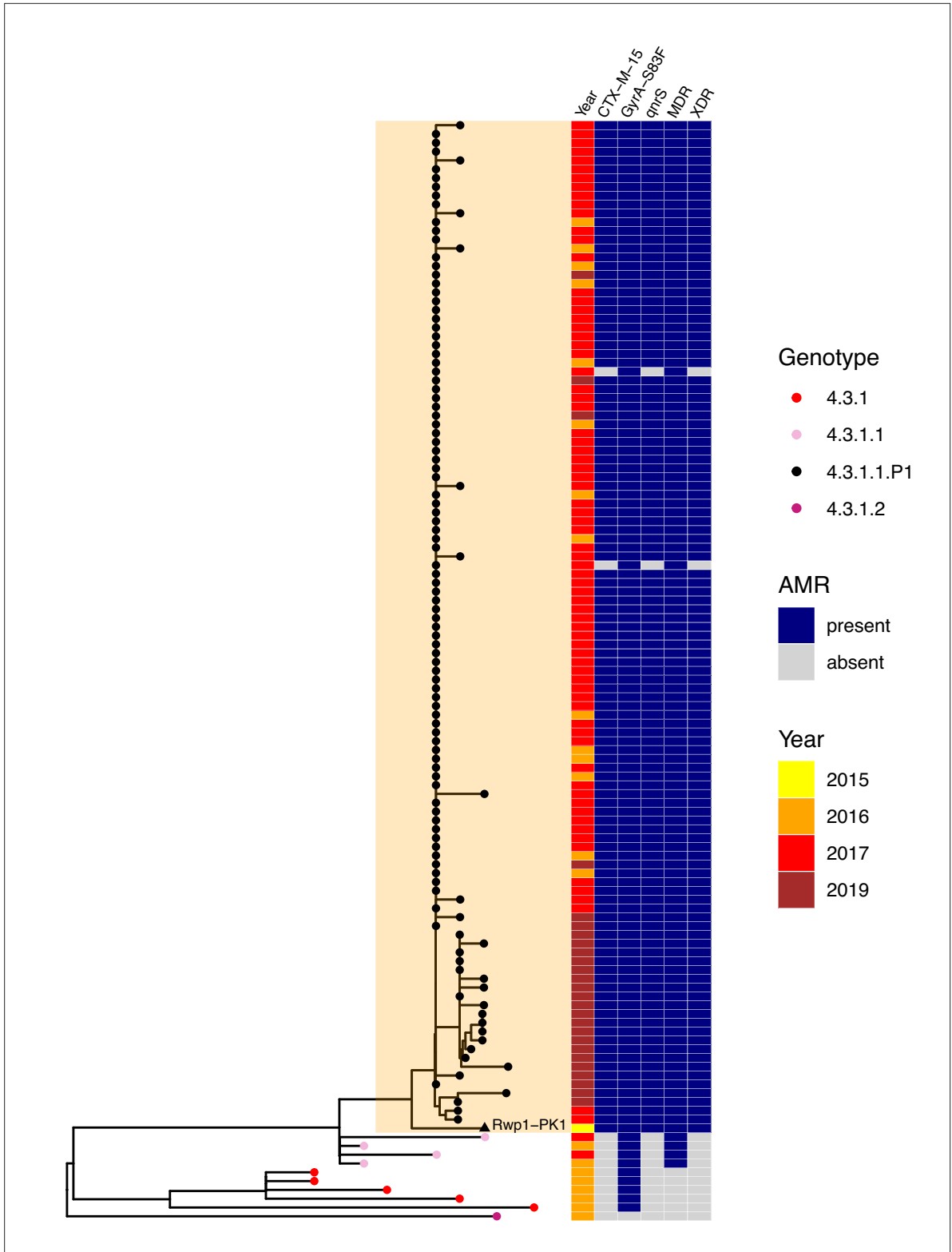

**Figure 4.** Phylogenetic tree showing position of 2015 Rwalpindi isolate, Rwp1-PK1, in context with other genomes from Pakistan. Core-genome distance-based neighbour-joining tree generated in Pathogenwatch, using all genomes from ***Klemm et al., 2018*** (the first genomic characterisation of the extensively drug-resistant [XDR] outbreak clade, including outbreak strains and local context strains from Sindh Province in 2016–2017) and ***Rasheed et al., 2020*** (genomic report of XDR outbreak strains from Lahore in 2019). Tree tips are coloured by genotype, according to inset legend; the 2015 strain Rwp1-PK1 is labelled in the tree and indicated with a triangle. Year of isolation and presence of antimicrobial resistance (AMR) determinants are indicated in the heatmap, according to inset legend.

**Table 3.** Extended spectrum beta-lactamase (ESBL) genes detected in Typhi genomes.

| Genotype | ESBL | 3GCR | Country of origin | n | Years | Other plasmid/AMR markers |
|---|---|---|---|---|---|---|
| 2.5.1 | CTX-M-15 | Y | DRC *Phoba et al., 2017* | 1 | 2015 | IncY[†]; $bla_{TEM-1}$, dfrA7, sul1 (gyrA-S83F) |
| | CTX-M-15 | Y | Philippines *Lagrada et al., 2022* | 1 | 2013 | – |
| 3 | SHV-12 | Y | Philippines *Hendriksen et al., 2015a*; *Lagrada et al., 2022* | 2 | 2007 | IncHI2A*; $bla_{TEM-1}$, dfrA18, tetA(D) |
| 3.3 | CTX-M-15 | Y | UK | 1 | 2012 | (gyrA-S83F) |
| 3.3.2 | CTX-M-15 | Y | Bangladesh *Djeghout et al., 2018*; *Tanmoy et al., 2018* | 2 | 2000 | IncI1*; $bla_{TEM-1}$ |
| 3.5 | CTX-M-12 | Y | Colombia *Guevara et al., 2021* | 1 | 2012 | IncL, IncFIB(pHCM2); $bla_{TEM-1}$, sul1 |
| | | Y | India *Sah et al., 2019* | 1 | 2019 | IncY*; qnrS, $bla_{TEM-1}$, dfrA14, sul2, (gyrA-S83Y) |
| 4.3.1 | CTX-M-15 | Y/N | Pakistan *da Silva et al., 2022*; *Klemm et al., 2018* | 2 | 2018 | IncY [‡]; qnrS, $bla_{TEM-1}$, sul2, catA1, dfrA7, sul1, tetA(A); (gyrA-S83F) |
| 4.3.1.1 | CTX-M-15 | N | Pakistan *da Silva et al., 2022* | 1 | 2016 | IncY; qnrS, $bla_{TEM-1}$, sul2, catA1, dfrA7, sul1, tetA(A); (gyrA-S83F) |
| | | Y | India *Klemm et al., 2018*; *Nair et al., 2021* | 1 | 2019 | |
| | | Y | Mexico https://wwwn.cdc.gov/narmsnow/ | 1 | 2019 | |
| | | Y | Pakistan *Klemm et al., 2018*; *Munir et al., 2016*; *Rasheed et al., 2020* | 656 | 2015–20 | |
| | CTX-M-15 | Y | USA https://wwwn.cdc.gov/narmsnow/ | 1 | 2019 | IncY*[†] qnrS, $bla_{TEM-1}$, sul2, catA1, dfrA7, sul1 |
| 4.3.1.1.P1 | CTX-M-55 | Y | Pakistan *Nair et al., 2021* | 1 | 2018 | (gyrA-S83F) |
| 4.3.1.2 | CTX-M-15 | Y | Iraq *Nair et al., 2021* | 2 | 2019 | IncY; (gyrA-S83F) |
| 4.3.1.2.1 | SHV-12 | Y | India *Argimón et al., 2021a*; *Chattaway et al., 2021*; *Ingle et al., 2021*; *Jacob et al., 2021* | 18 | 2015–20 | IncX3*; qnrB (gyrA-S83F, gyrA-D87N, parC-S80I) |

'Other plasmid/AMR markers' column includes: (i) plasmid replicons (Inc types) identified in the genome (in bold); (ii) other acquired AMR genes; (iii) chromosomal AMR mutations (in brackets). n.a. indicates susceptibility data not available.

*indicates this plasmid is the reported location of the ESBL gene in the genome assembly.

[†]n=31 4.3.1.1 .P1 isolates from Pakistan lacked plasmid replicons.

[‡]the ESBL 4.3.1. isolate from *Klemm et al., 2018*, was phenotypically third-generation cephalosporin resistant (3GCR), but the one from da Silva et al., 2022, was phenotypically 3GC sensitive.

Southern Asian countries were each represented by multiple in-country data sources plus travel-associated data collected in three or four other countries. *Figure 6—figure supplement 1* shows genotype prevalence estimates derived from these different sources (for laboratories contributing ≥20 isolates each) and *Figure 6a* shows the annual genotype frequency distributions (for years with ≥20 isolates). In most cases (67% of genotype-source combinations), genotype prevalence rates estimated from individual source laboratories yielded 95% CIs that overlapped with those of the pooled national estimates (see *Figure 6—figure supplement 1*). The main exception was for genotype 4.3.1.2 in India; for most source laboratories (many contributing via the Surveillance for Enteric Fever in India [SEFI] network; *Carey et al., 2020*; *da Silva et al., 2022*), this was the most prevalent genotype, but the point estimates ranged from 16% to 82%, compared with the pooled estimate of 53.4% (95% CI, 51.4–55.5%), and 95% CIs were frequently non-overlapping (see *Figure 6—figure supplement 1*). High prevalence of 4.3.1.2 was estimated from contributing laboratories in urban Vellore (82% [95% CI, 78–87%]), Chennai (67% [56–77%]), Bengaluru (70% [62–78%]), and Mumbai (two laboratories, estimates 74% [65–83%] and 63% [46–79%]); with lower prevalence in northern India, New Delhi (three laboratories, estimates 48% [28–68%], 40% [31–49%], 39% [22–56%]) and Chandigarh (39% [33–45%]). Two Indian laboratories were clear outliers, with little or no 4.3.1.2 but very high prevalence of a different genotype: 4.3.1.1 in rural Bathalapalli (81% [67–95%]) and 2.5 in the northern city of Ludhiana (77% [66–88%]). The relative prevalence of 4.3.1.1.P1 (XDR lineage) in Pakistan versus its parent lineage 4.3.1.1 also varied between sources, which could be explained by differences in the sampling periods and locations relative to the emergence of 4.3.1.1.P1 (see *Figure 6a*); notably, the highest estimate of XDR prevalence (n=27/27, 100%) came from a hospital-based study (*Rasheed*

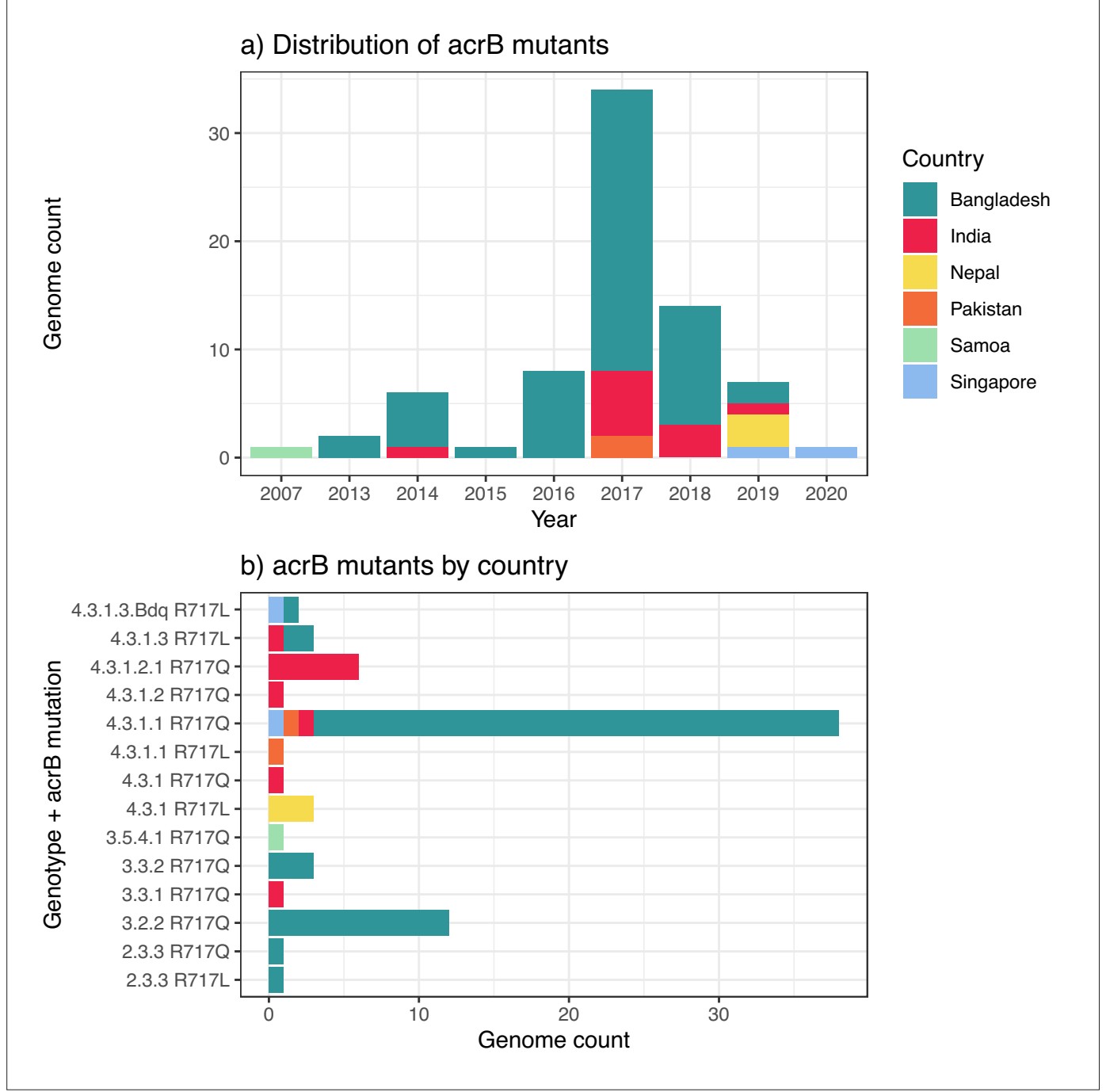

**Figure 5.** Distribution of azithromycin resistance-associated *acrB* mutations detected in Typhi genomes. (**a**) Temporal distribution of *acrB* mutants. (**b**) Distribution of *acrB* mutants by genotype and mutation. The first *acrB* mutant appeared in Samoa in 2007. Other mutants have appeared independently across a range of genetic backgrounds, largely in South Asian countries, but remain at low prevalence levels overall (see *Figure 2*). Country of origin is coloured as per inset label.

et al., 2020), which may select for more severe cases that were unresponsive to antibiotics received in the community setting. AMR prevalence estimates were also highly concordant across data sources (see *Figure 6—figure supplement 2*), and showed strikingly similar temporal trends (*Figure 6b*).

The only other country represented by ≥10 sequenced isolates each from multiple laboratories was Nigeria; these were located in Abuja (Zankli Medical Center, n=105, 2010–2013) and Ibadan (University of Ibadan, n=14, 2017–2018), and reference laboratories in England (n=15, 2015–2019) and the

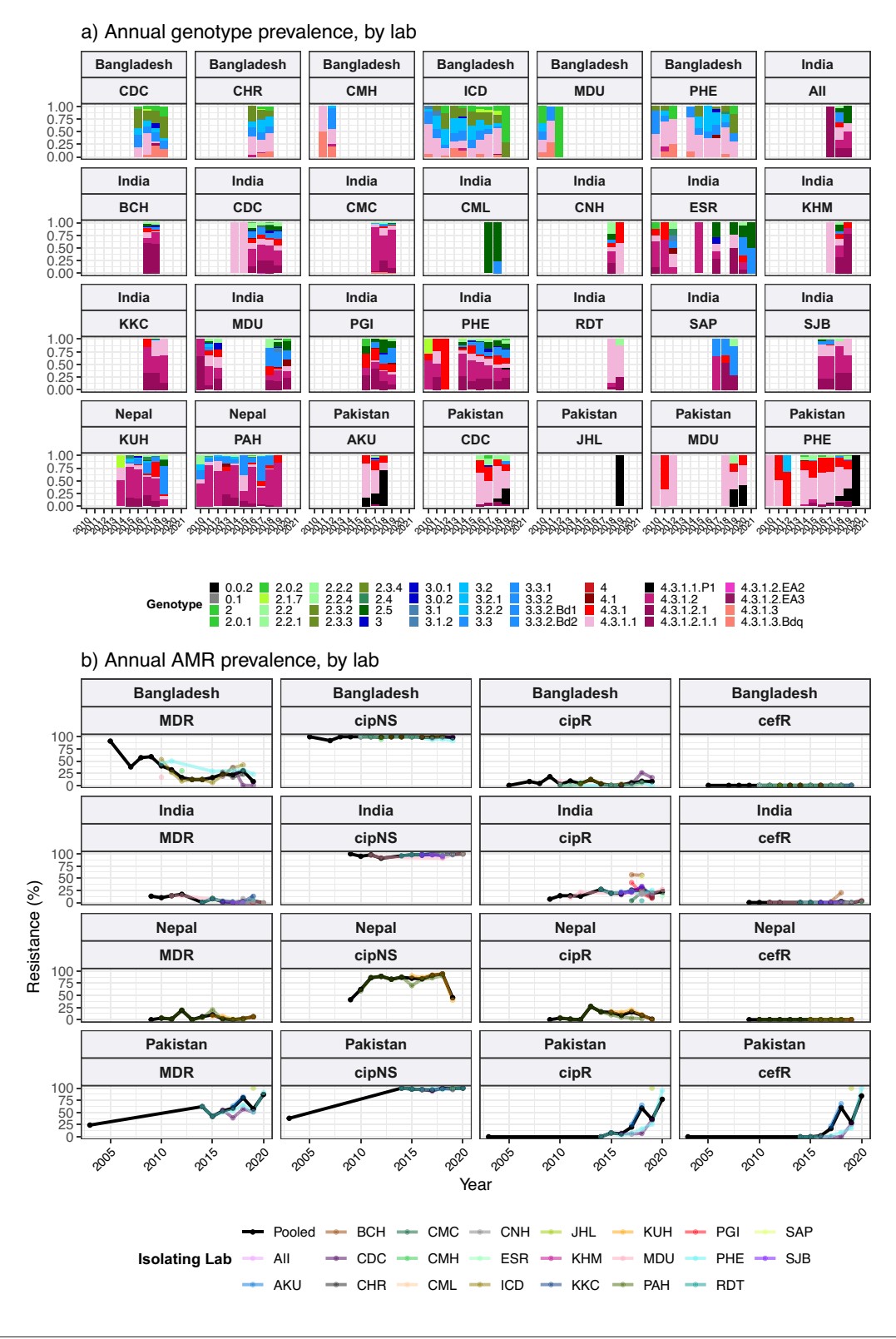

**Figure 6.** Annual genotype and antimicrobial resistance (AMR) frequencies by isolating lab, for South Asian countries with multiple data sources. Labs shown are those with ≥20 isolates; and years shown for each lab are those with N≥5 isolates from that year. (a) Bars are coloured to indicate annual genotype prevalence, as per inset legend. (b) Lines indicate annual frequencies of key AMR profiles, coloured by isolating laboratory as per inset

*Figure 6 continued on next page*

*Figure 6 continued*

legend. MDR, multidrug resistant; XDR, extensively drug resistant; CipNS, ciprofloxacin non-susceptible; CipR, ciprofloxacin resistant; CefR, ceftriaxone resistant. See *Supplementary file 9* for three-letter laboratory code master list.

The online version of this article includes the following figure supplement(s) for figure 6:

**Figure supplement 1.** Genotype prevalence estimated from different data sources, for South Asian countries.

**Figure supplement 2.** Antimicrobial resistance (AMR) prevalence estimated from different sources, for South Asian countries.

USA (n=10, 2016–2019) (see *Figure 7*). Genotype prevalence estimates were concordant across different sources, with single-laboratory 95% CIs overlapping with one another and with the pooled point estimate, for all five common genotypes (see *Figure 7*). The exception was that genotype 3.1.1 accounted for all n=14/14 isolates sequenced from Ibadan, but ranged from 53% to 70% prevalence at other laboratories and yielded a pooled national prevalence estimate of 67% [95% CI, 60–75%] (see *Figure 7a and c*). AMR prevalence estimates for Nigeria were more variable across laboratories (see *Figure 7b*), but this could be explained by their non-overlapping sampling times: Abuja data from earlier years (2010–2013) showed high MDR (49%) and low CipNS (4%); whereas Ibadan data from later years (2017–2018) showed comparatively lower MDR (21%) and higher CipNS (79%), in agreement with contemporaneous travel data (12% MDR, 60% CipNS, from total n=25 isolated 2015–2019).

## Discussion
### Strengths and limitations

This study presents the most comprehensive genomic snapshot of Typhi to date, with 12,965 high-quality genomes originating from 110 countries in 21 world regions. The consortium model provides improved consistency and completeness of source data aggregated from 77 laboratories and 66 unique studies. Our dataset also includes 1290 novel genomes sequenced by public health laboratories that would not otherwise have been published, including travel data from countries not previously represented in published Typhi genomics studies (e.g. El Salvador, Guatemala, Haiti, Mexico, and Peru). However, it is a post hoc analysis of isolates that were cultured in different contexts (including routine diagnostics, as well as study settings where culture would not normally be undertaken) and sequenced for different reasons (including retrospective studies, outbreak investigations, and routine surveillance). The study therefore has important limitations, most notably the scarcity of genomic data from many countries and world regions where typhoid is believed to be endemic (*GBD 2019 Antimicrobial Resistance Collaborators, 2022*), including Northern and Middle Africa, Western Asia, as well as Central and South America (*Figures 1–3*, *Figure 1—figure supplements 2–4*, *Figure 2—figure supplements 1–3*, *Figure 3—figure supplement 1*). These genomic data gaps reflect an underlying lack of routine blood culture or sustained blood-culture surveillance, and limited resources and expertise in many settings (*Ikhimiukor et al., 2022b*; *Iskandar et al., 2021*). In addition, public health authorities may be disincentivized to generate, analyse, and publish genomic data; we hope that this analysis strengthens the case for data generation and sharing for public good. Substantial investments have been made in recent years to improve and expand microbiological surveillance capacity in some low- and middle-income countries, but major regional surveillance gaps remain. It is therefore important to maximise information recovery from available data sources, especially WGS, which provides data on the emergence and spread of AMR variants. While the inference of AMR phenotype from WGS is currently highly reproducible and accurate for Typhi (*Argimón et al., 2021b*; *Chattaway et al., 2021*), continued phenotypic antimicrobial susceptibility testing remains crucial to monitor for emerging mechanisms and to guide changes in empiric therapy.

For now, routine sequencing of travel-associated Typhi infections diagnosed in high-income countries helps to fill some molecular surveillance gaps for some regions, assuming that accurate travel history is available and the sequence and metadata (including country of origin) are shared (*Ingle et al., 2019*). For example, our study included >3000 genomes shared by public health reference laboratories in England, Australia, New Zealand, France, Japan, and the USA. These infections mostly originate in other countries, and can in principle provide informative, if informal, sentinel surveillance for pathogen populations in countries with strong travel and/or immigration links to those with routine sequencing (*Ingle et al., 2019*). Indeed, for some countries and regions, travel data represented

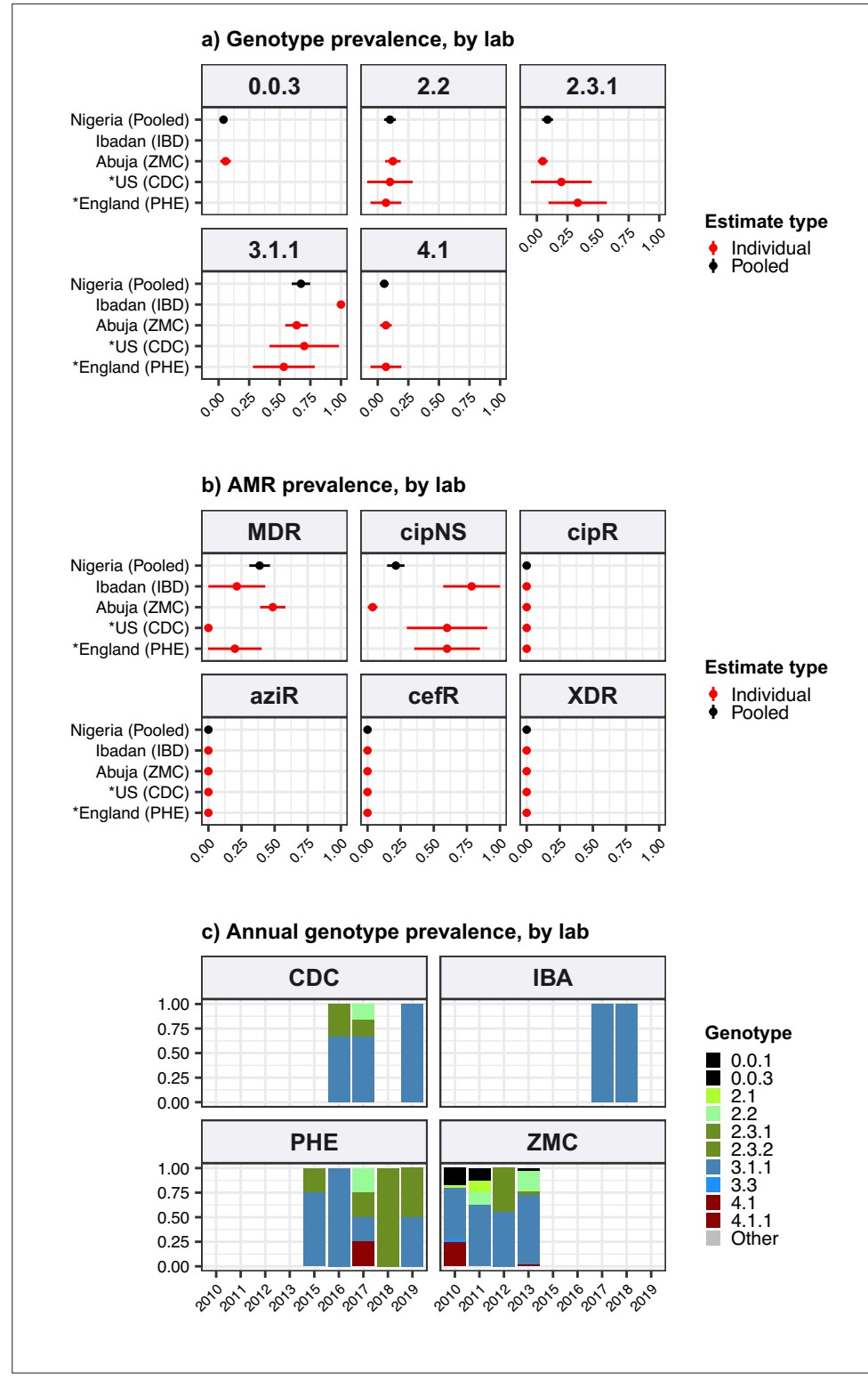

**Figure 7.** Genotype and antimicrobial resistance (AMR) prevalence rates estimated for Nigeria from different data sources. Data are shown only for source labs with N≥10 isolates from which to estimate prevalence. (**a**) Genotype prevalence and (**b**) AMR prevalence, using all available isolates per lab, 2010–2020. Lines show 95% confidence interval for each proportion (prevalence) estimate. Red indicates estimates based on data from individual labs, black indicates pooled estimates (i.e. from all labs), as per inset legend. (**c**) Annual genotype frequencies. Bars are coloured by genotype as per inset legend. Lab abbreviations are shown in y-axis labels for panels (**a**–**b**). MDR,

*Figure 7 continued on next page*

*Figure 7 continued*

multidrug resistant; XDR, extensively drug resistant; CipNS, ciprofloxacin non-susceptible; CipR, ciprofloxacin resistant; CefR, ceftriaxone resistant; AziR, azithromycin resistant. See ***Supplementary file 9*** for three-letter laboratory code master list.

most or all of the available genome data (see *Table 2*, ***Supplementary file 4***). In this study, where multiple data sources were available for the same country, we found that national genotype and AMR prevalence estimates for the period 2010–2020 were largely concordant between local surveillance studies and travel-associated cases captured elsewhere (***Figures 6–7***, ***Figure 6—figure supplements 1–2***), particularly when comparing contemporaneous annual prevalence estimates (***Figures 6 and 7c***). This clearly shows that travel-associated Typhi isolated in low burden countries can be informative for surveillance of some high burden countries, which should serve as incentive for public health reference laboratories to share their data to the fullest extent they are able to under local regulations and encourage culture-based diagnostics in those countries that rely primarily on clinical diagnosis of typhoid fever in local populations, and development of molecular diagnostic tests for local use as travel-associated infections may provide information on predominant genes encoding resistance.

Another key limitation stemming from the post hoc nature of this study is that it is hard to assess how representative the prevalence estimates are for a given region/country and timeframe. The GTGC has developed new source/metadata standards for Typhi (see Methods), that include information on the purpose of sampling, which were completed by the original owners of each dataset (data available in ***Supplementary file 2***). Such 'purpose-of-sampling' fields are currently lacking from metadata templates used for submission of bacterial genomes to the public sequencing archives (e.g. NCBI, ENA), and our approach was modelled on that established for sharing of SARS-CoV-2 sequence data, designed by the PHA4GE consortium (***Griffiths et al., 2022***). In this study, the purpose-of-sampling information was used to identify the subset of genome data that could be reasonably considered to be representative of national annual trends in genotype and AMR prevalence for public health surveillance purposes (n=9478 genomes post 2010; ***Figures 1–3***). These originate mainly from local typhoid surveillance studies (59%), or routine diagnostics/surveillance capturing locally acquired (19%) and travel-associated (24%) infections. The comparisons of estimates for a given country based on different sources of genomes (***Figures 6–7***, ***Figure 6—figure supplements 1–2***) are reassuring that the general scale and trends of AMR prevalence are reliable. The genome-based estimates are also in broad agreement with available phenotypic prevalence data on AMR in Typhi (***Browne et al., 2020***; ***Kariuki et al., 2015***), although systematic aggregation of susceptibility data is limited. Both phenotypic and genomic analyses necessarily reflect blood-culture-confirmed cases, which may be biased towards more resistant infections resulting in overestimation of AMR prevalence. Notably, the genome data adds an additional layer of information on resistance mechanisms and the emergence and spread of lineages or variants. Importantly, our study clearly shows that, whilst much attention has been given to the emergence and spread of drug-resistant H58 Typhi, other clones predominate outside of Southern Asia and Eastern Africa (***Figure 1***) and can be associated with CipNS (***Figure 2—figure supplements 3–4***), azithromycin (***Figure 6***), or ceftriaxone (***Table 3***), the drugs currently recommended by the World Health Organization as first choice treatment for enteric fever (***World Health Organization, 2022***).

## AMR

Our data demonstrate that CipNS is emerging or established in all regions except Melanesia (here represented by n=35 genomes from Fiji and Papua New Guinea, mainly from 2010, although more recent reports support a lack of CipNS in Fiji [***Davies et al., 2022***; ***Getahun Strobel et al., 2019***]; see ***Figure 2—figure supplement 1***). For countries with sufficient data to assess (≥50 genomes), CipNS was emerging or established in all countries except Ghana (***Figure 2***, ***Figure 3—figure supplement 1***), with no evidence of declining prevalence (***Figure 3***, ***Figure 3—figure supplement 1***). A diverse range of genotypes and QRDR mutations are involved (***Figure 2—figure supplements 3–4***), likely reflecting the lack of fitness cost associated with these mutations (***Baker et al., 2013***). That QRDR mutations are so widespread is highly concerning, as infections with CipNS strains can take longer to resolve, and full clinical resistance can emerge relatively easily against this background, through acquisition of either a mobile *qnr* gene (as occurred in 4.3.1.1.P1 in Pakistan) or additional QRDR mutations (as occurred in 4.3.1.2.1 in India). Notably, the data suggest that CipR typhoid is now a

well-established problem across Southern Asia and is emergent in Chile, Mexico, and South Africa (*Figures 2 and 3*, *Figure 2—figure supplement 1*, *Figure 3—figure supplement 1*). A recent study estimating national annual antibiotic consumption highlighted differences in rates of fluoroquinolone usage between regions and countries, which could potentially drive these differences in resistance prevalence (*Browne et al., 2021*). The highest rates of fluoroquinolone consumption were estimated in South Asian countries, rising from 1.67 defined daily doses (DDD) per 1000 per day in 2000 to 2.81 DDD/1000/day in 2010 and 2.94 DDD/1000/day in 2018 (see https://www.tropicalmedicine.ox. ac.uk/research/oxford/microbe/gram-project/antibiotic-usage-and-consumption). Fluoroquinolone consumption was also estimated to increase substantially in Latin America, rising from 0.64 DDD/1000/ day in 2000 to 1.85 DDD/1000/day in 2010 and 2.26 DDD/1000/day in 2018. Our data show that the highest incidence of CipR burden is associated with four main variants (*Figure 2—figure supplement 5*). In Pakistan, India, and Bangladesh, it is associated with locally emerged variants; however, the relatively high burden in Nepal is associated with variants acquired from India (*Britto et al., 2018*; *Pham Thanh et al., 2016a*). In other regions, CipR burden is low and so far linked mainly to the spread of 4.3.1.2.1 (*Britto et al., 2020*; *da Silva et al., 2022*) out of India (*Britto et al., 2020*; *da Silva et al., 2022*), plus occasional de novo emergence of resistant variants, which show no evidence of geographical spread (*Figure 2—figure supplement 5*). However, the high rates of CipNS in Kenya (53%) and Nigeria (40%) are concerning, especially given the increasing usage of fluoroquinolones in these countries (estimated 2.1 DDD/1000/day in 2018 in Kenya and 2.76 DD/1000/day in Nigeria) (*Browne et al., 2021*), which could potentially drive local emergence and spread of CipR.

While resistance to azithromycin and ceftriaxone have been detected (*Table 3*, *Figures 4–6b*, *Figure 2—figure supplements 1–2*, *Figure 3—figure supplement 1*), their prevalence remains low and, with the exception of XDR 4.3.1.1.P1, clonal expansion of resistant variants has not been observed. To our knowledge, there are no data reported on the fitness cost of *acrB* mutations or CefR plasmids in Typhi; however, the genomic evidence suggests a higher fitness cost compared with QRDR mutations, providing further support for the use of ceftriaxone or azithromycin over cipro-floxacin as we work to introduce preventative measures. Most instances of ESBL-gene carriage in Typhi (conferring CefR phenotype) have been short-lived (*Table 3*), suggesting selection against the acquisition of new ESBL genes or plasmids. The expansion and dominance of the XDR 4.3.1.1.P1 genotype in Pakistan is obviously concerning (*Figures 4 and 6*, *Figure 2—figure supplement 3a*, *Figure 3—figure supplement 1*); however, despite circulating at high prevalence in Pakistan for more than 5 years, the strain remains azithromycin-susceptible. There is also limited evidence of local trans-mission of 4.3.1.1.P1 in other countries; however, most countries near Pakistan have limited data available. A short local outbreak of XDR 4.3.1.1.P1 was reported in China, linked to contamination of an apartment block's water (*Wang et al., 2022*) and non-travel-associated cases have been reported in the USA (*Hughes et al., 2021*). Notably, a CefR+CipR lineage of 4.3.1.2.1 that appears to be well established in Mumbai, India, has been isolated only occasionally since 2015 (*Argimón et al., 2021b*; *Chattaway et al., 2021*; *Ingle et al., 2021*; *Jacob et al., 2021*; *Table 3*); however, this is the only example of persistence of a CefR strain besides 4.3.1.1.P1, and there is no evidence it has yet spread outside Mumbai. We hypothesise that the lack of widespread dissemination of 4.3.1.1.P1 and ESBL-positive 4.3.1.2.1 so far may be due to the fitness cost imposed by the associated plasmids (~85 Kbp IncY plasmid in 4.3.1.1.P1 [*Klemm et al., 2018*]; ~43 Kbp IncX3 plasmid in 4.3.1.2.1 [*Argimón et al., 2021b*]). The temporal trend data on MDR prevalence and IncHI1 plasmids (*Figure 3—figure supplement 2*) suggest that migration of the MDR locus from the plasmid to the chromosome may have mitigated the fitness cost associated with plasmid-borne MDR. The same may be true for ESBL genes, that is, the movement of the ESBL locus from the plasmid to the chromosome (as has recently been reported in 4.3.1.1.P1; *Nair et al., 2021*) may result in a fitter CefR or XDR variant that can spread more easily. Our data show that *acrB* mutations are occurring spontaneously and independently in multiple locations across a variety of genetic backgrounds (*Figure 5*). While they are still not preva-lent, increased use of azithromycin through public health programmes (e.g. trachoma elimination) as well as widespread misuse of azithromycin to treat SARS-CoV-2 infections and use of azithromycin as first-line therapy for typhoid-like illness may lead to increased selection pressure. It will therefore be important to maintain and expand genomic surveillance, particularly in typhoid endemic countries where azithromycin is used widely. It is also notable that, while they are rare overall, *acrB* mutations have already arisen in two of the most common CipR lineages (4.3.1.2 and 4.3.1.3.Bdq); this relatively

frequent co-occurrence warrants continued monitoring and investigation. While we did not detect the mobile AziR gene *mphA*, it is circulating in other *S. enterica* serovars (*Nair et al., 2016*; *Tack et al., 2022*) and other enteric bacteria that share plasmids with Typhi (including the human-specific *Shigella*; *Baker et al., 2018*), providing another potential mechanism for emergence of AziR in Typhi.

## Applications of genomic surveillance for typhoid fever control

We are at a pivotal stage in the history of typhoid control. Wider access to clean water and improved sanitation have led to a major reduction in global incidence of typhoid fever, which has also been reflected in declining incidence of other enteric diseases (*Steele et al., 2016*). This should continue but will require sustained investment from national and local governments and thus remains a long-term objective. In the short to medium term, widespread use of TCVs can help to further reduce global incidence of typhoid fever. The WHO has prequalified two TCVs and recommended their use in endemic countries, as well as settings where a high prevalence of AMR Typhi has been reported (*World Health Organization, 2018*). Gavi, the Vaccine Alliance, has committed funds to support the procurement and distribution of TCVs in typhoid endemic countries (*Gavi: The Vaccine Alliance, 2023a*; *Gavi: The Vaccine Alliance, 2023b*). Five countries have undertaken Gavi-supported national introductions (Pakistan, Liberia, Zimbabwe, Nepal, Malawi) and one country has self-financed a national introduction (Samoa) (*Neuzil, 2020*; *Sikorski, 2020*). In Pakistan and Zimbabwe, TCV introduction was stimulated by the occurrence of AMR Typhi outbreaks in major urban centres, highlighting that the case for prevention can be stronger when curative therapy is less available. Additional support is likely required to inform TCV decision-making in other typhoid endemic countries, particularly where burden and AMR data are scarce.

With increasingly limited treatment options, vaccines are an even more important tool to mitigate the public health burden of AMR Typhi, both through the prevention of drug-resistant infections and through broader, indirect effects, like reduction of empiric antimicrobial use leading to reduced selection pressure. While TCVs have been shown to be highly effective against drug-resistant Typhi (*Batool et al., 2021*; *Yousafzai et al., 2021*), public health policymakers have to weigh the value of TCVs against other competing immunisation priorities. While TCV introduction is scaled up globally, antimicrobial stewardship should also be prioritised. Aggregated, representative data showing distribution and temporal trends in AMR can inform local treatment guidelines to extend the useful lifespan of antimicrobials licensed to treat typhoid fever, potentially including reverting to former last-line drugs in some settings. The traffic light system presented in this analysis (see *Figure 2* and *Figure 2—figure supplement 1*) provides a framework for monitoring trends in AMR and adjusting empiric therapy guidelines accordingly. The WHO recently released its AWaRe (Access, Watch, Reserve) treatment guidelines (*World Health Organization, 2022*), which indicate that choice of empiric therapy should be guided by severity of presentation and local risk of fluoroquinolone resistance; if low risk, oral ciprofloxacin is recommended for both mild and severe cases and if there is a high risk of fluoroquinolone resistance, oral azithromycin is recommended for mild cases and intravenous ceftriaxone is recommended for severe cases. However, the guidelines do not indicate which prevalence rate of resistance should warrant avoidance of treatment with ciprofloxacin, nor do they indicate where high prevalence rates of resistance might be expected, although it is noted that drug resistance is most prevalent in Asia. There is an opportunity to further refine these recommendations with additional, local information about AMR prevalence and trends over time. Additional data are required from resource-limited settings, where typhoid fever diagnosis is often based on clinical presentation, to optimise these recommendations.

Genomic surveillance has a particularly important role to play in monitoring for changes in clinically important resistances in Typhi, as a shift in resistance mechanism or early evidence of clonal spread, which can only be identified definitively using WGS, could provide early warning of a likely increase in prevalence. This study provides an analytical framework for Typhi genomic analysis, based on an open, robust, reproducible data flow and analysis framework leveraging open-access online data analysis platforms (Typhi Mykrobe for read-based genotyping [*Ingle et al., 2022*]; the GHRU pipeline for genome assembly [*Underwood, 2020*], and Typhi Pathogenwatch for assembly-based genotyping and tree-building [*Argimón et al., 2021b*]). We have made available all data processing and statistical analysis code, and underlying sequence and metadata, via GitHub and FigShare (see Methods). Together, these provide (i) a comprehensive data and code resource for the research and

public health communities interested in typhoid surveillance data; (ii) a model for the inclusion of WGS in project-based or routine surveillance studies of typhoid that can be readily replicated and adapted; and (iii) a sustainable model for aggregated analysis of typhoid genomic surveillance data that can readily incorporate new data and extract features (genotypes, AMR determinants, plasmid replicons) of importance to clinical and public health audiences. Notably, this consortium-driven effort shows that new insights can be gained from aggregated analysis of published data, which were not evident from the individual contributing studies, for example (i) the XDR strain 4.3.1.1.P1 existed in Pakistan in 2015, a year earlier than previously reported (*Figure 4*); (ii) the CefR+CipR strain reported in Mumbai (*Argimón et al., 2021b*; *Jacob et al., 2021*) has persisted between at least 2015 and 2020 and is now more easily identified as 4.3.1.2.1 with $bla_{SHV-12}$; (iii) persistence of MDR in certain settings is correlated with migration of MDR from plasmid to chromosome (*Figure 3—figure supplement 2*), which has implications for the future persistence and potentially spread of ESBL strains.

This dataset provides clear, actionable information about the distribution and temporal trends in AMR across multiple countries and regions. Where data gaps exist, the potential of travel-associated data to serve as 'sentinel' surveillance has been demonstrated previously by *Ingle et al., 2019*, and supported by additional data included in this analysis. These data can and should inform prioritisation of TCV introduction and improvements to WASH infrastructure. Sustaining and expanding genomic surveillance can also facilitate measuring the impact of TCV introduction on local bacterial populations, as has been done for previous vaccines like pneumococcal conjugate vaccines. In addition, monitoring for potential 'strain replacement' with other *Salmonella* serovars following TCV introduction can and should inform the prioritisation of the development and deployment of future combination *Salmonella* vaccines.

The SARS-CoV-2 pandemic illustrated the power of open, continuous data sharing and crowd-sourced analysis, and the importance of ensuring that genomic surveillance leads to local benefits. The scale of this analysis, which was made possible through the efforts of an extensive network of collaborators, enables the extraction of key insights of public health relevance. The authors hope that this consortium effort serves as a starting point for continued data generation and sharing and collective analysis, with additional participation from an expanded group of stakeholders. In particular, we hope that researchers and public health authorities from areas with little publicly available data see the value of reporting and sharing genomic data for collective public health benefit. In addition, we hope that the current momentum for donor and government support of molecular surveillance is sustained, so that additional groups are able to generate their own data and fill regional data gaps to inform local public health action.

## Additional information

### Group author details

**Global Typhoid Genomics Consortium Group Authorship**
**Peter Aaby**: Bandim Health Project, Guinea-Bissau, Guinea-Bissau; **Ali Abbas**: Department of Microbiology, Faculty of Veterinary Medicine, University of Kufa, Najaf, Iraq; **Niyaz Ahmed**: University of Hyderabad, Hyderabad, India; **Saadia Andleeb**: Atta-ur-Rahman School of Applied Biosciences, National University of Sciences and Technology, Islamabad, Pakistan; **Abraham Aseffa**: Armauer Hansen Research Institute, Addis Ababa, Ethiopia; **Kate S Baker**: University of Liverpool, Liverpool, United Kingdom; **Adwoa Bentsi-Enchill**: World Health Organization, Geneva, Switzerland; **Robert F Breiman**: Emory University, Atlanta, United States; **Carl Britto**: Boston Children's Hospital, Boston, United States; **Josefina Campos**: INEI-ANLIS "Dr Carlos G. Malbrán", Buenos Aires, Argentina; **Chih-Jun Chen**: Chang Gung Memorial Hospital, Taoyuan, Taiwan; **Chien-Shun Chiou**: Centers for Disease Control, Taipei, Taiwan; **Viengmon Davong**: Lao-Oxford-Mahosot Hospital-Wellcome Trust Research Unit (LOMWRU), Microbiology Laboratory, Mahosot Hospital, Vientiane, Lao People's Democratic Republic; **Abul Faiz**: Dev Care Foundation, Dhaka, Bangladesh; **Danish Gul**: Atta-ur-Rahman School of Applied Biosciences, National University of Sciences and Technology, Islamabad, Pakistan; **Rumina Hasan**: Aga Khan University, Karachi, Pakistan; **Mochammad Hatta**: Department of Molecular Biology and Immunology, Faculty of Medicine, Hasanuddin University, Makassar, Indonesia; **Aamer Ikram**: National Institute of Health, Islamabad, Pakistan; **Lupeoletalalelei Isaia**: Ministry of

Health, Government of Samoa, Apia, Samoa; **Jan Jacobs**: Department of Clinical Sciences, Institute of Tropical Medicine, Antwerp, Belgium; Department of Microbiology, Immunology and Transplantation, KU Leuven, Leuven, Belgium; **Simon Kariuki**: Malaria Branch, Kenya Medical Research Institute (KEMRI) Centre for Global Health Research, Kisumu, Kenya; **Fahad Khokhar**: Department of Veterinary Medicine, University of Cambridge, Cambridge, United Kingdom; **Elizabeth Klemm**: Novo Nordisk Foundation, Hellerup, Denmark; **Laura MF Kuijpers**: Department of Infectious Diseases, Leiden University Medical Center, Leiden, Netherlands; **Gemma Langridge**: Quadram Institute Bioscience, Norwich, United Kingdom; **Kruy Lim**: Sihanouk Hospital Center of HOPE, Phnom Penh, Cambodia; **Octavie Lunguya**: Department of Microbiology, Institut National de Recherche Biomédicale, Kinshasa, Democratic Republic of the Congo; Department of Medical Biology, University Teaching Hospital of Kinshasa, Kinshasa, Democratic Republic of the Congo; **Francisco Luquero**: Global Alliance for Vaccines and Immunization (GAVI), Geneva, Switzerland; **Calman A MacLennan**: Jenner Institute, Nuffield Department of Medicine, University of Oxford, Oxford, United Kingdom; **Florian Marks**: International Vaccine Institute, Seoul, Republic of Korea; Cambridge Institute of Therapeutic Immunology and Infectious Disease, University of Cambridge School of Clinical Medicine, Cambridge Biomedical Campus, Cambridge, United Kingdom; Madagascar Institute for Vaccine Research, University of Antananarivo, Antananarivo, Madagascar; Heidelberg Institute of Global Health, University of Heidelberg, Heidelberg, Germany; **Masatomo Morita**: National Institute of Infectious Diseases, Tokyo, Japan; **Mutinta Muchimba**: Centre of Infectious Disease Research in Zambia, Lusaka, Zambia; **James CL Mwansa**: Lusaka Apex Medical University, Lusaka, Zambia; **Kapambwe Mwape**: Centre of Infectious Disease Research in Zambia, Lusaka, Zambia; Water and Health Research Center, Faculty of Health Sciences, University of Johannesburg, Johannesburg, South Africa; Department of Basic Medical Sciences, Michael Chilufya Sata School of Medicine, Copperbelt University, Ndola, Zambia; **Jason M Mwenda**: World Health Organization (WHO) Regional Office for Africa, Immunization and Vaccines Development, Brazzaville, Democratic Republic of the Congo; **John Nash**: National Microbiology Laboratory, Public Health Agency of Canada, Toronto, Canada; **Kathleen M Neuzil**: Center for Vaccine Development and Global Health, University of Maryland School of Medicine, Baltimore, Baltimore, United States; **Paul Newton**: Lao-Oxford-Mahosot Hospital-Wellcome Trust Research Unit (LOMWRU), Microbiology Laboratory, Mahosot Hospital, Vientiane, Lao People's Democratic Republic; Centre for Tropical Medicine & Global Health, Nuffield Department of Medicine, University of Oxford, Oxford, United Kingdom; **Stephen Obaro**: University of Nebraska Medical Center, Omaha, United States; International Foundation Against Infectious Diseases in Nigeria, Abuja, Nigeria; **Sophie Octavia**: Environmental Health Institute, National Environment Agency, Singapore, Singapore; **Makoto Ohnishi**: National Institute of Infectious Diseases, Tokyo, Japan; **Michael Owusu**: Department of Medical Diagnostics, Kwame Nkrumah University of Science and Technology, Kumasi, Ghana; **Ellis Owusu-Dabo**: School of Public Health, Kwame Nkrumah University of Science and Technology, Kumasi, Ghana; **Se Eun Park**: International Vaccine Institute, Seoul, Republic of Korea; Yonsei University Graduate School of Public Health, Seoul, Republic of Korea; **Julian Parkhill**: Department of Veterinary Medicine, University of Cambridge, Cambridge, United Kingdom; **Duy Thanh Pham**: Oxford University Clinical Research Unit, Ho Chi Minh, Viet Nam; **Marie-France Phoba**: Department of Microbiology, Institut National de Recherche Biomédicale, Kinshasa, Democratic Republic of the Congo; **Derek J Pickard**: Department of Medicine, University of Cambridge, Cambridge, United Kingdom; **Raphael Rakotozandrindrainy**: Madagascar Institute for Vaccine Research, University of Antananarivo, Antananarivo, Madagascar; **Pilar Ramon-Pardo**: Pan American Health Organization, AMR Special Program, Washington, United Kingdom; **Farhan Rasheed**: Allama Iqbal Medical College, Lahore, Pakistan; **Assaf Rokney**: Ministry of Health, Jerusalem, Israel; **Priscilla Rupali**: Department of Infectious Diseases, Christian Medical College, Vellore, India; **Ranjit Sah**: Tribhuvan University Teaching Hospital, Institute of Medicine, Kathmandu, Nepal; **Sadia Shakoor**: Pediatrics and Child Health, Aga Khan University, Karachi, Pakistan; London School of Hygiene & Tropical Medicine, London, United Kingdom; **Michelo Simuyand**: Center of Infectious Disease Research in Zambia, Lusaka, Zambia; **Arvinda Sooka**: National Institute for Communicable Diseases, Johannesburg, South Africa; **Jeffrey D Stanaway**: Institute for Health Metrics and Evaluation, University of Washingto, Washington, United States; **A Duncan Steele**: Enteric and Diarrheal Diseases, Bill & Melinda Gates Foundation, Seattle, United States; **Bieke Tack**: Department of Microbiology, Immunology and Transplantation, KU Leuven, Leuven, Belgium; **Adama Tall**: Institut Pasteur, Dakar, Senegal; **Neelam Taneja**: Department of Medical

Microbiology, Postgraduate Institute of Medical Education and Research, Chandigarh, India; **Mekonnen Teferi**: Armauer Hansen Research Institute, Addis Ababa, Ethiopia; **Sofonias Tessema**: Africa Centres for Disease Prevention and Control, Addis Ababa, Ethiopia; **Gaetan Thilliez**: Quadram Institute Bioscience, Norwich, United Kingdom; **Paul Turner**: Cambodia Oxford Medical Research Unit, Angkor Hospital for Children, Siem Reap, Cambodia; **James E Ussher**: University of Otago, Dunedin, New Zealand; **Annavi Marie Villanueva**: National Reference Laboratory for HIV/AIDS, Hepatitis, and Other Sexually-Transmitted Infections, San Lazaro Hospital, Manila, Philippines; **Bart Weimer**: Department of Population Health and Reproduction, 100K Pathogen Genome Consortium, School of Veterinary Medicine, UC Davis, Davis, United States; **Vanessa K Wong**: Addenbrooke's Hospital, Cambridge University Hospitals NHS Foundation Trust, Cambridge Biomedical Campus, Cambridge, United Kingdom; **Raspail Carrel Founou**: Department of Microbiology, Immunology and Haematology, Faculty of Medicine and Pharmaceutical Sciences, University of Dschang, Dschang, Cameroon; Antimicrobial Resistance and Infectious Diseases Unit, Research Institute of Centre of Expertise and Biological Diagnostic of Cameroon (CEDBCAM-RI), Dschang, Cameroon

## Competing interests

Nicholas A Feasey: NAF chairs the Wellcome Surveillance and Epidemiology of Drug Resistant Infections (SEDRIC) group, which has a focus on antimicrobial resistance. This could be perceived as relevant although not a direct conflict. Isaac I Bogoch: IB has consulted to BlueDot and the NHL Players' Association. Andrew J Pollard: AJP is chair of the UK Department of Health and Social Care's (DHSC) Joint Committee on Vaccination and Immunisation (JCVI) but does not take part in the JCVI COVID-19 committee. He was a member of WHO SAGE until 2022. AJPs employer, Oxford University has entered into a partnership with AstraZeneca for development of a COVID-19 vaccine. AJP has provided advice to Shionogi & Co., Ltd on development of a COVID19 vaccine. The other authors declare that no competing interests exist.

## Funding

| Funder | Grant reference number | Author |
|---|---|---|
| HORIZON EUROPE Marie Sklodowska-Curie Actions | 845681 | Zoe A Dyson |
| Bill and Melinda Gates Foundation | OPP1217121 | Robert A Kingsley |
| Biotechnology and Biological Sciences Research Council | BB/R012504/1 and BBS/E/F/00PR10348 | Tapfumanei Mashe |
| Wellcome Trust | Senior Fellowship | Stephen Baker |
| Bill and Melinda Gates Foundation | OPP1194582 | Myron M Levine |
| National Institutes of Health | F30AI156973 | Michael J Sikorski |
| Bill and Melinda Gates Foundation | OPP1113007 | Jason R Andrews |
| Bill and Melinda Gates Foundation | INV-042340 | Senjuti Saha |
| Bill and Melinda Gates Foundation | OPPGH5231 | John A Crump |
| National Institutes of Health | U01AI062563 | John A Crump |
| National Institute for Health Research | 16_136_111 | Ravikumar Kadahalli Lingegowda Varun Shamanna |
| Wellcome Trust | 206194 | Ravikumar Kadahalli Lingegowda Varun Shamanna |

| Funder | Grant reference number | Author |
|---|---|---|
| Bill and Melinda Gates Foundation | OPP1020327 | Grant Austin Mackenzie |
| Canadian Institutes of Health Research | | Isaac I Bogoch |
| National Institute for Health Research | Global Health Unit on Genomic Surveillance of AMR | Iruka N Okeke |
| National Institute for Health Research | | Robert S Heyderman |
| Department of Health and Social Care | Fleming Fund | Anthony M Smith |
| UK Medical Research Council | MR/L00464X/ | Iruka N Okeke |
| Bill and Melinda Gates Foundation | INV-036234 | Iruka N Okeke |
| National Institutes of Health | R01AI099525 | Samuel Kariuki |
| Institut Pasteur and Santé Publique France | | Elisabeth Njamkepo François-Xavier Weill |
| Medical Research Council | Joint Global Health Trials Scheme MR/TOO5033/1 | Christopher M Parry |
| Bill and Melinda Gates Foundation | | Andrew R Greenhill Roy Robins-Browne Jivan Shakya |
| United States Department of Health and Human Services | U19AI110820 | David A Rasko |
| Indian Council of Medical Research | | Arti Kapil |
| World Health Organization and Gavi, the Vaccine Alliance | | Andrew J Pollard |
| National Institute for Health and Care Research | NIHR Professor of Global Health | Nicholas A Feasey |
| Department for Health and Social Care, the Department for International Development/Global Challenges Research Fund, the UK Medical Research Council, and the Wellcome Trust | | Christopher M Parry |
| National Institute for Health Research | National Institute for Health Research Health Protection Research Unit (NIHR HPRU) in Genomics and Enabling Data at University of Warwick in partnership with the UK Health Security Agency (UKHSA) | Marie Anne Chattaway |
| National Institutes of Health | R01TW009237 | John A Crump |
| National Institutes of Health | R01AI121378 | John A Crump |

| Funder | Grant reference number | Author |
|---|---|---|
| Bill and Melinda Gates Foundation | OPP1558210 | John A Crump |
| Bill and Melinda Gates Foundation | OPP1151153 | John A Crump<br>Melita A Gordon |
| Bill and Melinda Gates Foundation | INV-008335 | Jason R Andrews |
| Bill and Melinda Gates Foundation | INV-029806 | Myron M Levine |
| Bill and Melinda Gates Foundation | OPP1161058 | Myron M Levine |
| Bill & Melinda Gates Foundation | INV-108979 | Siaosi Tupua |
| Wellcome | | Karnika Saigal |
| Bill & Melinda Gates Foundation | OPP1141321 | Melita A Gordon |
| Wellcome | 206545/7/17/Z and 106158/7/14/Z | Melita A Gordon |
| National Institute for Health Research | Professorship NIHR300039 | Melita A Gordon |
| National Institute for Health Research | NIHR300039 | Philip M Ashton |
| Bill & Melinda Gates Foundation | OPP1175797 | Kathryn E Holt |

The funders had no role in study design, data collection and interpretation, or the decision to submit the work for publication. For the purpose of Open Access, the authors have applied a CC BY public copyright license to any Author Accepted Manuscript version arising from this submission.

## Author contributions

Megan E Carey, Conceptualization, Data curation, Formal analysis, Validation, Investigation, Visualization, Writing – original draft, Project administration, Writing – review and editing; Zoe A Dyson, Data curation, Formal analysis, Investigation, Visualization, Methodology, Project administration, Writing – review and editing; Danielle J Ingle, Formal analysis, Investigation, Visualization, Writing – original draft, Writing – review and editing; Afreenish Amir, Mabel K Aworh, Ka Lip Chew, Christopher M Parry, Formal analysis, Visualization, Writing – review and editing; Marie Anne Chattaway, John A Crump, Nicholas A Feasey, Benjamin P Howden, Karen H Keddy, Mailis Maes, Sandra Van Puyvelde, Hattie E Webb, Formal analysis, Investigation, Visualization, Writing – review and editing; Ayorinde Oluwatobiloba Afolayan, Anna P Alexander, Shalini Anandan, Jason R Andrews, Philip M Ashton, Buddha Basnyat, Ashish Bavdekar, Isaac I Bogoch, John D Clemens, Kesia Esther da Silva, Anuradha De, Joep de Ligt, Paula Lucia Diaz Guevara, Christiane Dolecek, Shanta Dutta, Marthie M Ehlers, Louise Francois Watkins, Denise O Garrett, Gauri Godbole, Melita A Gordon, Andrew R Greenhill, Chelsey Griffin, Madhu Gupta, Rene S Hendriksen, Robert S Heyderman, Yogesh Hooda, Juan Carlos Hormazabal, Odion O Ikhimiukor, Junaid Iqbal, Jobin John Jacob, Claire Jenkins, Dasaratha Ramaiah Jinka, Jacob John, Gagandeep Kang, Abdoulie Kanteh, Arti Kapil, Abhilasha Karkey, Samuel Kariuki, Robert A Kingsley, Roshine Mary Koshy, AC Lauer, Myron M Levine, Ravikumar Kadahalli Lingegowda, Stephen P Luby, Grant Austin Mackenzie, Tapfumanei Mashe, Chisomo Msefula, Ankur Mutreja, Geetha Nagaraj, Savitha Nagaraj, Satheesh Nair, Take K Naseri, Susana Nimarota-Brown, Elisabeth Njamkepo, Iruka N Okeke, Sulochana Putli Bai Perumal, Andrew J Pollard, Agila Kumari Pragasam, Firdausi Qadri, Farah N Qamar, Sadia Isfat Ara Rahman, Savitra Devi Rambocus, David A Rasko, Pallab Ray, Roy Robins-Browne, Temsunaro Rongsen-Chandola, Jean Pierre Rutanga, Samir K Saha, Senjuti Saha, Karnika Saigal, Mohammad Saiful Islam Sajib, Jessica C Seidman, Jivan Shakya, Varun Shamanna, Jayanthi Shastri, Rajeev Shrestha, Sonia Sia, Michael J Sikorski, Ashita Singh, Anthony M Smith, Kaitlin A Tagg, Dipesh Tamrakar, Arif Mohammed Tanmoy, Maria Thomas, Mathew S Thomas, Robert Thomsen,

Siaosi Tupua, Krista Vaidya, Mary Valcanis, Balaji Veeraraghavan, François-Xavier Weill, Jackie Wright, Gordon Dougan, Investigation, Writing – review and editing; Nicholas R Thomson, Resources, Investigation, Writing – review and editing; Silvia Argimón, Data curation, Methodology, Writing – review and editing; Jacqueline A Keane, Data curation, Writing – review and editing; David M Aanensen, Investigation, Methodology, Project administration, Writing – review and editing; Stephen Baker, Conceptualization, Investigation, Writing – original draft, Project administration, Writing – review and editing; Kathryn E Holt, Conceptualization, Data curation, Formal analysis, Investigation, Visualization, Methodology, Writing – original draft, Project administration, Writing – review and editing; Global Typhoid Genomics Consortium Group Authorship, Writing – review and editing

**Author ORCIDs**
Megan E Carey  https://orcid.org/0000-0002-7797-9080
Zoe A Dyson  http://orcid.org/0000-0002-8887-3492
John A Crump  https://orcid.org/0000-0002-4529-102X
Benjamin P Howden  http://orcid.org/0000-0003-0237-1473
Christopher M Parry  https://orcid.org/0000-0001-7563-7282
Sandra Van Puyvelde  http://orcid.org/0000-0001-8434-5732
Hattie E Webb  https://orcid.org/0000-0002-1190-7930
Ayorinde Oluwatobiloba Afolayan  http://orcid.org/0000-0003-1405-2365
Jason R Andrews  http://orcid.org/0000-0002-5967-251X
Anuradha De  http://orcid.org/0000-0002-8701-4100
Paula Lucia Diaz Guevara  http://orcid.org/0000-0002-8252-5622
Louise Francois Watkins  http://orcid.org/0000-0002-1165-1162
Odion O Ikhimiukor  https://orcid.org/0000-0002-3738-4584
Junaid Iqbal  http://orcid.org/0000-0002-5014-2176
Dasaratha Ramaiah Jinka  http://orcid.org/0000-0002-8184-279X
Gagandeep Kang  http://orcid.org/0000-0002-3656-564X
Samuel Kariuki  http://orcid.org/0000-0003-3209-9503
Chisomo Msefula  http://orcid.org/0000-0003-2304-886X
Satheesh Nair  https://orcid.org/0000-0002-7297-1485
Iruka N Okeke  https://orcid.org/0000-0002-1694-7587
Samir K Saha  http://orcid.org/0000-0003-3820-0748
Senjuti Saha  http://orcid.org/0000-0001-6087-6766
David M Aanensen  http://orcid.org/0000-0001-6688-0854
Kathryn E Holt  http://orcid.org/0000-0003-3949-2471

**Decision letter and Author response**
Decision letter https://doi.org/10.7554/eLife.85867.sa1
Author response https://doi.org/10.7554/eLife.85867.sa2

---

## Additional files

**Supplementary files**
• Supplementary file 1. Details of local ethical approvals provided for studies that were unpublished at the time of contributing data to this consortium project. Most data are now published, and the citations for the original studies are provided here. National surveillance programs in Chile (*Maes et al., 2022*), Colombia (*Guevara et al., 2021*), France, New Zealand, and Nigeria (*Ikhimiukor et al., 2022b*) were exempt from local ethical approvals as these countries allow sharing of non-identifiable pathogen sequence data for surveillance purposes. The US CDC Internal Review Board confirmed their approval was not required for use in this project (#NCEZID-ARLT-10/20/21-fa687).

• Supplementary file 2. Line list of 13,000 genomes included in the study.

• Supplementary file 3. Source information recorded for genomes included in the study. ^Indicates cases included in the definition of 'assumed acute illness'.

• Supplementary file 4. Summary of genomes by country.

• Supplementary file 5. Genotype frequencies per region (N, %, 95% confidence interval; annual and aggregated, 2010–2020).

• Supplementary file 6. Genotype frequencies per country (N, %, 95% confidence interval; annual and aggregated, 2010–2020).

- Supplementary file 7. Antimicrobial resistance (AMR) frequencies per region (N, %, 95% confidence interval; aggregated 2010–2020).
- Supplementary file 8. Antimicrobial resistance (AMR) frequencies per country (N, %, 95% confidence interval; annual and aggregated, 2010–2020).
- Supplementary file 9. Laboratory code master list. Three letter laboratory codes assigned by the consortium.
- MDAR checklist

### Data availability

All data analysed during this study are publicly accessible. Raw Illumina sequence reads have been submitted to the European Nucleotide Archive (ENA), and individual sequence accession numbers are listed in *Supplementary file 2*. The full set of n=13,000 genome assemblies generated for this study are available for download from FigShare: https://doi.org/10.26180/21431883. All assemblies of suitable quality (n=12,849) are included as public data in the online platform Pathogenwatch (https://pathogen.watch). The data are organised into collections, which each comprise a neighbour-joining phylogeny annotated with metadata, genotype, AMR determinants, and a linked map. Each contributing study has its own collection, browsable at https://pathogen.watch/collections/all?organismId=90370. In addition, we have provided three large collections, each representing roughly a third of the total dataset presented in this study: Typhi 4.3.1.1 (https://pathogen.watch/collection/2b7mp173dd57-clade-4311), Typhi lineage 4 (excluding 4.3.1.1) (https://pathogen.watch/collection/wgn6bp1c8bh6-clade-4-excluding-4311), and Typhi lineages 0-3 (https://pathogen.watch/collection/9o4bpn0418n3-clades-0-1-2-and-3). In addition, users can browse the full set of Typhi genomes in Pathogenwatch and select subsets of interest (e.g. by country, genotype, and/or resistance) to generate a collection including neighbour-joining tree for interactive exploration.

The following dataset was generated:

| Author(s) | Year | Dataset title | Dataset URL | Database and Identifier |
|---|---|---|---|---|
| Holt K | 2023 | Whole genome assemblies of *Salmonella enterica* serovar Typhi assembled for the Global Typhoid Genomics Consortium paper: "Global diversity and antimicrobial resistance of typhoid fever pathogens: insights from 13,000 *Salmonella* Typhi genomes" | https://doi.org/10.26180/21431883 | figshare, 10.26180/21431883 |

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
