## [Editor Report]

Although largely descriptive, this meta-analysis of 13,000 published *Salmonella* Typhi (Typhi) genomes is very important to public health. The dataset and presented analysis are convincing, representing the first wholesale analysis of all available Typhi genomes from the last 21 years. The findings are of interest to microbiologists and infectious disease physicians as well as to public health epidemiologists and policy makers, as they are of great significance to tracking the emergence and maintenance of AMR in Typhi and include novel insights into XDR strain emergence in Pakistan, as well as the relationship between MDR maintenance and chromosomal integration.

---

## [Decision Letter]

**Decision letter after peer review:**

Thank you for submitting your article "Global diversity and antimicrobial resistance of typhoid fever pathogens: insights from a meta-analysis of 13,000 *Salmonella* Typhi genomes" for consideration by *eLife*. Your article has been reviewed by 3 peer reviewers, including Marc J Bonten as the Reviewing Editor and Reviewer #1, and the evaluation has been overseen by Jos van der Meer as the Senior Editor. The following individual involved in review of your submission has agreed to reveal their identity: Lauren A Cowley (Reviewer #2).

Essential revisions:

1) Reviewer #2 has several recommendations for improving clarity in reporting.

2) Reviewer #3 has questions and concerns related to the generalisability of the findings due to the potential bias in the individual studies. The authors address this aspect in the discussion, but I would like to ask them to determine whether the feel more emphasis is needed.

*Reviewer #2 (Recommendations for the authors):*

I have just a few recommendations for the study that I think would be useful to readers:

1) Although the dataset is particularly large and difficult to represent phylogenetically, I think a tree is still an important and necessary output for this dataset collection. A lot of readers will want to look at the phylogeography in a more classical representation than what is available in Figure 1. This doesn't need to be of all 13,000 strains and could be a subset that represents the major lineages in different countries. It might also be helpful to have a neighbor-joining tree (as you used in your smaller tree figures) for the whole dataset on microreact. I realise that doing a ML tree for the whole dataset would take considerable compute time so think that a neighbor joining tree would be adequate for a quick representation for reader browsing on microreact. Readers may then want to select particular clades from that which they go on to download the genomes from and generate a ML tree for.

2) A table that details empiric therapy guidelines and potential regional variations in that would be useful to go alongside the AMR section. Clinicians and public health policy makers will be interested in the direct relationship between the therapy guidelines and the AMR trends observed in this meta-analysis.

3) I noticed that a few of the tables and figures still had PHE instead of UKHSA, this should be updated.

4) The supplementary figure and table download point was not obvious to me. I suggest adding in a clear sentence in the Data availablity section that states that the supplementary figures and tables are available from the Zenodo link.

*Reviewer #3 (Recommendations for the authors):*

While we commend the authors for the volume of work undertaken to establish the consortium and analyze the data, there are major concerns on the validity of the findings due to the inconsistent sampling and potential lack of representativeness. Using the findings in this paper to fundamentally change the field of *Salmonella* Typhi genomics requires a more rigorous assessment of the representativeness and generalizability of the data. While these data are extremely useful in studying the biology and evolution of *Salmonella* Typhi, they are not adequate for making general statements on the distribution of genotypes in multiple regions/ countries. Although the authors recognized the limitations of using an aggregation of studies that lack a consistent sampling strategy, they have drawn general inferences on the prevalence of genotypes in the absence of a rigorous approach to mitigate sampling bias.

Nonetheless there is an opportunity to study the evolution of key lineages such as the H58 genotypes in detail. What is the phylogenetic topology of these lineages? Is there phylogenetic clustering based on Country? Perhaps the authors could explore a phylogenetic dating approach to reconstruct the evolutionary history of this lineage. Additionally, how are the sub lineages evolving? Are they adapting in a country specific manner, or is the diversity simply due to founder effects and clonal inheritance? For example, when did the 4.3.1.3 lineage emerge in Bangladesh? Are there any genomic signatures, either in the core or accessory genome, that are unique to this lineage, which might explain localization.

---

## [Author Response]

Essential revisions:1) Reviewer #2 has several recommendations for improving clarity in reporting.

Please see detailed responses below.

2) Reviewer #3 has questions and concerns related to the generalisability of the findings due to the potential bias in the individual studies. The authors address this aspect in the discussion, but I would like to ask them to determine whether the feel more emphasis is needed.

We have taken on board these concerns and have addressed them by clarifying how biased sampling frames were identified and removed before calculating prevalence estimates and expanding discussion of these issues.

Reviewer #2 (Recommendations for the authors):I have just a few recommendations for the study that I think would be useful to readers:1) Although the dataset is particularly large and difficult to represent phylogenetically, I think a tree is still an important and necessary output for this dataset collection. A lot of readers will want to look at the phylogeography in a more classical representation than what is available in Figure 1. This doesn't need to be of all 13,000 strains and could be a subset that represents the major lineages in different countries. It might also be helpful to have a neighbor-joining tree (as you used in your smaller tree figures) for the whole dataset on microreact. I realise that doing a ML tree for the whole dataset would take considerable compute time so think that a neighbor joining tree would be adequate for a quick representation for reader browsing on microreact. Readers may then want to select particular clades from that which they go on to download the genomes from and generate a ML tree for.

We appreciate the reviewer’s suggestion of creating one large neighbor joining tree to facilitate exploration of the dataset by readers. In discussions with the Pathogenwatch team, it was determined that a single tree would be too large and difficult to navigate, so we have instead generated three collections of 4-5k genomes each, organised by genotype that readers can browse/explore:

– Collection 1: Typhi 4.3.1.1; n=4,121 genomes (available: https://pathogen.watch/collection/2b7mp173dd57-clade-4311)

– Collection 2: Typhi lineage 4 (excluding 4.3.1.1); n=3,553 genomes (available: https://pathogen.watch/collection/wgn6bp1c8bh6-clade-4-excluding-4311)

– Collection 3: Typhi lineages 0-3; n=5,171 genomes (available: https://pathogen.watch/collection/9o4bpn0418n3-clades-0-1-2-and-3)

Also note that since all the Typhi genomes are publicly available in Pathogenwatch, it is possible for users to select subsets of interest (e.g. by country, genotype, and/or resistance) and construct a neighbour-joining tree for their chosen subset.

We have added the following text to the Data Availability section, to alert readers to these options for viewing / generating phylogenies:

All assemblies of suitable quality (n=12,849) are included as public data in the online platform Pathogenwatch (https://pathogen.watch). The data are organised into collections, which each comprise a neighbour-joining phylogeny annotated with metadata, genotype, AMR determinants, and a linked map. Each contributing study has its own collection, browsable at https://pathogen.watch/collections/all?organismId=90370. In addition, we have provided three large collections each representing roughly a third of the total dataset presented in this study: Typhi 4.3.1.1 (https://pathogen.watch/collection/2b7mp173dd57-clade-4311), Typhi lineage 4 (excluding 4.3.1.1) (https://pathogen.watch/collection/wgn6bp1c8bh6-clade-4-excluding-4311), and Typhi lineages 0-3 (https://pathogen.watch/collection/9o4bpn0418n3-clades-0-1-2-and-3). In addition, users can browse the full set of Typhi genomes in Pathogenwatch and select subsets of interest (e.g. by country, genotype, and/or resistance) to generate a collection including neighbour-joining tree for interactive exploration.

In addition, the Global Typhoid Genomics Consortium is currently working on a follow-up study exploring phylogenetic analyses of these data. We initially established two working groups for analysis – one focused on descriptive statistics and epidemiology, and one focused on phylodynamic analysis. Given the volume of work to conduct and results to present, we ultimately decided to split the outputs of these two working groups into two separate manuscripts; the one currently under consideration, and a second manuscript that provides detailed phylogenetic analyses including spatiotemporal phylodynamics. The second manuscript is still in preparation, and we plan to preprint it in the coming months and submit to *eLife* as a Research Advance building on the current manuscript.

2) A table that details empiric therapy guidelines and potential regional variations in that would be useful to go alongside the AMR section. Clinicians and public health policy makers will be interested in the direct relationship between the therapy guidelines and the AMR trends observed in this meta-analysis.

Unfortunately, we are not aware of any source of aggregated empiric therapy guidelines for typhoid fever; nor of adherence to guidelines. Each country (and in many countries, subregions) set their own guidelines and there is no collated source of these. We agree this could be an important resource, which we could attempt to tackle as a Consortium; however, this would constitute a separate study in itself.

The recently released WHO AWaRe antibiotic book (December 2022) sets out guidelines for therapy, which depend on the local risk of fluoroquinolone resistance; if low risk, oral ciprofloxacin is recommended for mild and severe cases and if high risk, then oral azithromycin is recommended for mild cases and intravenous ceftriaxone for severe cases. However, the guidelines do not set out what level of resistance should warrant avoidance of ciprofloxacin, nor where such levels are expected to be encountered, although it is noted that resistance is most common in Asia. We have added details of these new guidelines to the Applications of genomic surveillance for typhoid fever control subsection of the Discussion section:

“The WHO recently released its AWaRe (Access, Watch, Reserve) treatment guidelines, which indicate that choice of empiric therapy should be guided by severity of presentation and local risk of fluoroquinolone resistance; if low risk, oral ciprofloxacin is recommended for both mild and severe cases and if there is a high risk of fluoroquinolone resistance, oral azithromycin is recommended for mild cases and intravenous ceftriaxone is recommended for severe cases. However, the guidelines do not indicate which prevalence rate of resistance should warrant avoidance of treatment with ciprofloxacin, nor do they indicate where high prevalence rates of resistance might be expected, although it is noted that drug-resistance is most prevalent in Asia. There is an opportunity to further refine these recommendations with additional, local information about AMR prevalence and trends over time. Additional data are required from resource-limited settings, where typhoid fever diagnosis is often based on clinical presentation, to optimise these recommendations.”

3) I noticed that a few of the tables and figures still had PHE instead of UKHSA, this should be updated.

Thank you for this comment! During the data gathering phase, we developed a Laboratory Code Master List and assigned three-digit lab codes to institutions that were contributing data. These are listed in Table S9. We maintained “PHE” in some of the figures, as it refers to the three-digit lab code that was assigned to the data that came from PHE publications, but we have updated the PHE line listing in Table S9 to read “United Kingdom Health Security Agency (formerly Public Health England, PHE)”

4) The supplementary figure and table download point was not obvious to me. I suggest adding in a clear sentence in the Data availablity section that states that the supplementary figures and tables are available from the Zenodo link.

Many thanks for this feedback, and we agree that this could be made clearer! We have added additional language to the Data Availability section indicating that all metadata and supplementary figures and tables may be accessed using the Zenodo link and we have ensured that all supplementary figures and tables have also been uploaded in the full submission.

Reviewer #3 (Recommendations for the authors):While we commend the authors for the volume of work undertaken to establish the consortium and analyze the data, there are major concerns on the validity of the findings due to the inconsistent sampling and potential lack of representativeness. Using the findings in this paper to fundamentally change the field of *Salmonella* Typhi genomics requires a more rigorous assessment of the representativeness and generalizability of the data. While these data are extremely useful in studying the biology and evolution of Salmonella Typhi, they are not adequate for making general statements on the distribution of genotypes in multiple regions/ countries. Although the authors recognized the limitations of using an aggregation of studies that lack a consistent sampling strategy, they have drawn general inferences on the prevalence of genotypes in the absence of a rigorous approach to mitigate sampling bias.Nonetheless there is an opportunity to study the evolution of key lineages such as the H58 genotypes in detail. What is the phylogenetic topology of these lineages? Is there phylogenetic clustering based on Country? Perhaps the authors could explore a phylogenetic dating approach to reconstruct the evolutionary history of this lineage. Additionally, how are the sub lineages evolving? Are they adapting in a country specific manner, or is the diversity simply due to founder effects and clonal inheritance? For example, when did the 4.3.1.3 lineage emerge in Bangladesh? Are there any genomic signatures, either in the core or accessory genome, that are unique to this lineage, which might explain localization.

We thank the reviewer for their comments and feedback.

1. It appears that we have not made clear enough one of the unique features of our data aggregation effort, which was to ask data generators to share more complete metadata on their genome collections, including ‘purpose of sampling’, so that we could identify and remove sets of genomes resulting from ‘targeted’ sampling frames such as outbreak investigations or sequencing of AMR cases. Whilst our total data set includes 13,000 genomes, the prevalence estimates reported are estimated from the subset of n=9,478 genomes that met the stated criteria as stated in Methods section “Genotype and AMR prevalence estimates and statistical analysis”:

“Inclusion criteria for these estimates were: known ‘country of origin’, known year of isolation, non-targeted sampling, assumed acute illness (see definitions of these variables above). A total of 10,726 genomes met these criteria; the subset of 9,478 isolated from 2010 onwards were the focus of the majority of analyses and visualisations, including all prevalence estimates. The prevalence estimates reported in text and figures are simple proportions; 95% confidence intervals for proportions are given in text and supplementary tables where relevant. Annual prevalences were estimated for countries that had N≥50 representative genomes and ≥3 years with ≥10 representative genomes.”

Definitions are given in the Metadata curation and variable definitions subsection of the Methods section:

“To identify isolate collections that were suitably representative of local pathogen populations, for the purpose of calculating genotype and AMR prevalences for a given setting, data owners were asked to indicate the purpose of sampling for each study or dataset. Options available were either ‘Non Targeted’ (surveillance study, routine diagnostics, reference lab, other; n=11,086), ‘Targeted’ (cluster investigation, AMR focused, other; n=1,862) or ‘Not Provided’ (n=17).”

We have also added at the end of this section, *“*Only samples from “Non Targeted” sampling frames with known year of isolation and country of origin were included in national prevalence estimates” to further clarify that our national prevalence estimates do not include samples isolated using targeted sampling frames.

The Results section Overview of available data introduces our approach to prevalence estimates as follows:

“In total, n=10,726 genomes were assumed to represent acute typhoid fever and recorded as derived from ‘non-targeted’ sampling frames, i.e. local population-based surveillance studies or reference laboratory-based national surveillance programmes that could be considered representative of a given time (year of isolation) and geography (country and region of origin) (see Methods for definitions). The majority of these isolates (n=9,478, 88.4%) originate from 2010 onwards, hence we focus our reporting of genotype and AMR prevalences on this period. Most come from local typhoid surveillance studies (n=5,574) or routine diagnostics/reference laboratory referrals capturing locally acquired (n=1,543) or travel-associated (n=2,284) cases. All prevalence estimates reported in this study derive from this data subset, unless otherwise stated.”

To make it clearer to readers, we have added the phrase "from non-targeted sampling frames” explicitly to the beginning of the “Geographic distribution of genotypes” section. We note the “Global distribution of AMR” Results section already begins “We estimated the regional and national prevalence of clinically relevant AMR profiles in Typhi for the period 2010-2020, inferred from WGS data from non-targeted sampling frames for which country of origin could be determined (as per genotype prevalences, see Methods).”

We have also expanded the opening of the “Robustness of national estimates across studies” Results section to highlight that the data presented comes from the subset of studies:

“The estimates of genotype and AMR prevalence represented here reflect post hoc analyses of data that were generated for a variety of different primary purposes in different settings, by different groups using varied criteria for sample collection, including in-country surveillance and travel-associated cases recorded in other countries. Whilst data sets known to be biased towards sequencing of AMR strains including outbreak investigations were excluded from prevalence estimates, there is still substantial heterogeneity across data sources. To explore the robustness of these national-level estimates, we compared prevalence estimates for the same country from different studies/sources, where sufficient data existed to do so.”

2. In relation to the specific question about data sources from Pakistan; Table 1 of the manuscript lists the data sources, and column 3 of the table indicates the number of genomes from each study that are included in prevalence estimates. As Table 1 shows, the genomes sequenced in Klemm et al., 2018 (i.e. a targeted investigation of an XDR Typhi outbreak) were excluded from prevalence estimates. Yousafzai et al., 2019 is not a source of genome data and so does not contribute to our AMR/genotype prevalence estimates (the paper was cited as a reference for regional spread of XDR within Pakistan, however). The n=27 genomes from Rasheed et al. 2019 were included (as shown in Table 1); whilst we agree with the reviewer that Typhi isolated in hospitals may be biased towards AMR strains, in many settings, blood culture is rarely conducted outside hospitals and so excluding such data is not feasible. We have added a note to the Results section where estimates of 4.3.1.1.P1 (XDR) prevalence in Pakistan are compared from different sources:

“notably, the highest estimate of XDR prevalence (n=27/27, 100%) came from a hospital-based study (Rasheed et al., 2019), which may select for more severe cases that were unresponsive to antibiotics received in the community setting.”

We note also that this is a general problem with estimating AMR rates for Typhi, using genomics or phenotyping, as a culture step is a pre-requisite that may well bias samples towards more severe cases and/or those more likely to be drug resistant. E.g. for most countries, this will be the case with Typhi data reported to WHO GLASS also. We have added explicit mention of this issue in the Discussion:

“The genome-based estimates are also in broad agreement with available phenotypic prevalence data on AMR in Typhi (Browne et al., 2020; Kariuki et al., 2015), although systematic aggregation of susceptibility data are limited. Both phenotypic and genomic analyses necessarily reflect blood-culture-confirmed cases, which may be biased towards more resistant infections resulting in overestimation of AMR prevalence.”

The figure quoted in the abstract (“Extensively drug-resistant (XDR) typhoid has become dominant in Pakistan (70% in 2020)”) is indeed the prevalence estimate from our (filtered) data set. We take the point that without being able to provide sufficient context in the abstract, it is unwise to provide such specific estimates and we have modified this to simply read “Extensively drug-resistant (XDR) typhoid has become dominant in Pakistan”.

3. To explore robustness of national prevalences calculated from our heterogeneous source data, we undertook robustness analyses using data from five countries from which we had multiple in-country data sources as well as travel-associated data (Bangladesh, India, Nepal, Pakistan, Nigeria) to assess how 95% confidence intervals of genotype and AMR prevalence estimates from single sites within these countries compare to national ones. Relevant text from the Robustness of national estimates across studies subsection within the Results section:

“Southern Asian countries were each represented by multiple in-country data sources plus travel-associated data collected in three or four other countries. Figure S12 shows genotype prevalence estimates derived from these different sources (for laboratories contributing ≥20 isolates each) and Figure 6a shows the annual genotype frequency distributions (for years with ≥20 isolates). In most cases (67% of genotype-source combinations), genotype prevalences estimated from individual source laboratories yielded 95% confidence intervals (CIs) that overlapped with those of the pooled national estimates (see Figure S12). The main exception was for genotype 4.3.1.2 in India; for most source laboratories (many contributing via the Surveillance for Enteric Fever in India (SEFI) network (Carey et al., 2020; da Silva et al., 2022)), this was the most prevalent genotype, but the point estimates ranged from 16% to 82%, compared with the pooled estimate of 53.4% (95% CI, 51.4-55.5%), and 95% CIs were frequently non-overlapping (see Figure S12). High prevalence of 4.3.1.2 was estimated from contributing laboratories in urban Vellore (82% [95% CI, 78-87%]), Chennai (67% [56-77%]), Bengaluru (70% [62-78%]), and Mumbai (two laboratories, estimates 74% [65-83%] and 63% [46-79%]); with lower prevalence in northern India, New Delhi (three laboratories, estimates 48% [28-68%], 40% [31-49%], 39% [22-56%]) and Chandigarh (39% [33-45%]). Two Indian laboratories were clear outliers, with little or no 4.3.1.2 but very high prevalence of a different genotype: 4.3.1.1 in rural Bathalapalli (81% [67-95%]) and 2.5 in the northern city of Ludhiana (77% [66-88]). The relative prevalence of 4.3.1.1.P1 (XDR lineage) in Pakistan versus its parent lineage 4.3.1.1 also varied between sources, which could be explained by differences in the sampling periods and locations relative to the emergence of 4.3.1.1.P1 (see Figure 6a); notably, the highest estimate of XDR prevalence (n=27/27, 100%) came from a hospital-based study (Rasheed et al., 2019), which may select for more severe cases that were unresponsive to antibiotics received in the community setting. AMR prevalence estimates were also highly concordant across data sources (see Figure S13), and showed strikingly similar temporal trends (Figure 6b).

The only other country represented by ≥10 sequenced isolates each from multiple laboratories was Nigeria; these were located in Abuja (Zankli Medical Center, n=105, 2010-2013) and Ibadan (University of Ibadan, n=14, 2017-2018), and reference laboratories in England (n=15, 2015-2019) and the USA (n=10, 2016-2019) (see Figure 7). Genotype prevalence estimates were concordant across different sources, with single-laboratory 95% CIs overlapping with one another and with the pooled point estimate, for all five common genotypes (see Figure 7). The exception was that genotype 3.1.1 accounted for all n=14/14 isolates sequenced from Ibadan, but ranged from 53-70% prevalence at other laboratories and yielded a pooled national prevalence estimate of 67% [95% CI, 60-75%] (see Figure 7a, c). AMR prevalence estimates for Nigeria were more variable across laboratories (see Figure b), but this could be explained by their non-overlapping sampling times: Abuja data from early years (2010-2013) showed high MDR (49%) and low CipNS (4%); whereas Ibadan data from later years (2017-2018) showed comparatively lower MDR (21%) and higher CipNS (79%), in agreement with contemporaneous travel data (12% MDR, 60% CipNS, from total n=25 isolated 2015-2019).”

This approach obviously cannot eliminate all possible bias, and we discuss some of these limitations in the Strengths and limitations section of the Discussion section:

“Another key limitation stemming from the post hoc nature of this study is that it is hard to assess how representative the prevalence estimates are for a given region/country and timeframe. The GTGC has developed new source/metadata standards for Typhi (see Methods), that include information on the purpose of sampling, which were completed by the original owners of each dataset (data available in Table S2). Such ‘purpose-of-sampling’ fields are currently lacking from metadata templates used for submission of bacterial genomes to the public sequencing archives (e.g. NCBI, ENA), and our approach was modelled on that established for sharing of SARS-CoV-2 sequence data, designed by the PHA4GE consortium (Griffiths et al., 2022). In this study, the purpose-of-sampling information was used to identify the subset of genome data that could be reasonably considered to be representative of national annual trends in genotype and AMR prevalence for public health surveillance purposes (n=9,478 genomes post 2010; Figures1-3). These originate mainly from local typhoid surveillance studies (59%), or routine diagnostics/surveillance capturing locally acquired (19%) and travel-associated (24%) infections. The comparisons of estimates for a given country based on different sources of genomes (Figures6-7, S12-13) are reassuring that the general scale and trends of AMR prevalence are reliable. The genome-based estimates are also in broad agreement with available phenotypic prevalence data on AMR in Typhi (Browne et al., 2020; Kariuki et al., 2015), although systematic aggregation of susceptibility data are limited. Notably, the genome data adds an additional layer of information on resistance mechanisms and the emergence and spread of lineages or variants. Importantly, our study shows clearly that, whilst much attention has been given to the emergence and spread of drug-resistant H58 Typhi, other clones predominate outside of Southern Asia and Eastern Africa (Figure 1) and can be associated with non-susceptibility to ciprofloxacin (Figures S7, S9), azithromycin (Figure 6) or ceftriaxone (Table 3), which are included in the World Health Organization Essential Medicines List as first choice treatment for enteric fever (World Health Organization, 2019).”

We have added additional language acknowledging the reviewer’s point about the potential for both genomic and phenotypic data to be biased towards severity later in the Strengths and Limitations section:

“Both phenotypic and genomic analyses necessarily reflect blood-culture-confirmed cases, which may be biased towards more resistant infections resulting in overestimation of AMR prevalence.”

In light of the above, we feel that prevalence estimates generated using the “surveillance ready” subset of our data are reasonably representative of their respective countries of origin, and that these findings are presented in the context of possible limitations.

4. To the reviewer’s points about exploration of lineage evolution and phylogeographic analyses, these points will be explored in greater detail in a manuscript that is currently in development (see response to Reviewer 2), which we also aim to submit to *eLife* as a Research Advance following the current manuscript; the present work focuses more on an updated overview of global genotype and AMR distribution and trends over time.

5. Figure 4 (phylogeny of 4.3.1.1.1P1) was included to confirm that the 2015 strain Rwp1-PK1 is part of the 4.3.1.1.P1 XDR lineage. This strain was not included in the Klemm 2018 study and predates recognition of the XDR outbreak by a year. We feel this genome is important as an example of the power of aggregated genomic analysis (as noted by Reviewer 2) and of the genotyping nomenclature. It clearly shows that the XDR strain 4.3.1.1.P1 emerged at least a year earlier than previously thought, a fact that was missed by the Klemm 2018 studies and others, despite the case report being published in 2016 and the genome in 2017. This is because the XDR lineage did not yet have a genotype designation until after the Klemm 2018 report, and there were no mechanisms for systematic screening of public databases for related strains; hence the 2015 XDR strain Rwp1-PK1 XDR was not identified as part of the outbreak lineage until years later.

The tree is shown in Figure 4 to provide evidence that the strain Rwp1-PK1 (labelled in the figure) not only genotypes as 4.3.1.1.P1 using marker SNPs, but does in fact cluster with the rest of this lineage in the whole-genome phylogeny and also shares the same AMR determinants with the rest of the lineage.